# Small Singular Values Matter: A Random Matrix Analysis of Transformer Models

**Max Staats**
Center for Scalable Data Analytics and Artificial Intelligence
Leipzig University
staats@itp.uni-leipzig.de

**Matthias Thamm**
Institute for Theoretical Physics
Leipzig University
thamm@itp.uni-leipzig.de

**Bernd Rosenow**
Institute for Theoretical Physics
Leipzig University
rosenow@physik.uni-leipzig.de

## Abstract

This work analyzes singular-value spectra of weight matrices in pretrained transformer models to understand how information is stored at both ends of the spectrum. Using Random Matrix Theory (RMT) as a zero information hypothesis, we associate agreement with RMT as evidence of randomness and deviations as evidence for learning. Surprisingly, we observe pronounced departures from RMT not only among the largest singular values – the usual outliers – but also among the smallest ones. A comparison of the associated singular vectors with the eigenvectors of the activation covariance matrices shows that there is considerable overlap wherever RMT is violated. Thus, significant directions in the data are captured by small singular values and their vectors as well as by the large ones. We confirm this empirically: zeroing out the singular values that deviate from RMT raises language-model perplexity far more than removing values from the bulk, and after fine-tuning the smallest decile can be the third most influential part of the spectrum. To explain how vectors linked to small singular values can carry more information than those linked to larger values, we propose a linear random-matrix model. Our findings highlight the overlooked importance of the low end of the spectrum and provide theoretical and practical guidance for SVD-based pruning and compression of large language models.

## 1 Introduction

Large language models (LLMs) have become foundational in deep learning, revolutionizing natural language processing tasks such as translation, text classification, and question answering [27, 45, 48, 51]. Despite the well-documented success [29], a thorough theoretical understanding of their inner workings remains incomplete. Although researchers have investigated various facets of LLMs [37], fundamental questions persist about how these models encode information and the specific roles of their components.

One promising approach for gaining deeper insights is the application of random matrix theory (RMT), which has proved effective for identifying structural properties and information density in neural networks [31, 43, 44]. In particular, RMT analysis of the spectrum of weight matrices can help determine where information is located within models. When networks are randomly initialized, the weight distributions precisely matches RMT predictions. After training, deviations from these predictions reveal how model parameters have adapted.

39th Conference on Neural Information Processing Systems (NeurIPS 2025).

Building on these insights, we use RMT to pinpoint regions in LLMs where relevant features are encoded, by identifying deviations from the RMT-predicted spectrum. We study the singular value spectra of weight matrices from three pretrained models: Bert[1] [26], Pythia[2] [9], and Llama-8B[3] [14]. We identify the regions lying outside the theoretically predicted Marchenko-Pastur spectrum [30] as areas of feature learning by comparing the corresponding singular vectors with the covariance matrix of the layer activations, finding strong similarity. Interestingly, this phenomenon is not only present for the largest but also for the smallest singular values. This similarity stays consistent across different blocks of the transformer architectures and holds for all three models we examine. When removing groups of singular values (and associated vectors) from these models, performance degrades most significantly for the smallest and largest singular values that violate RMT properties.

Additionally, we contribute to the ongoing discussion about removing small singular values in LLMs. Previous studies suggest that small singular values can be relevant for generalization [23], while other work indicates potential benefits from removing them [41]. However, the most common perspective is that they are negligible, as their removal is the optimal low rank $W'$ solution for weights $W$ under the L2 norm $|W' - W|_2$ [15]. We reconcile these perspectives by showing in which matrix types small singular values are important, and that the potential damage that is done by removing the smallest singular values from a pretrained transformer can be recovered by a fine-tuning step. Our results are of crucial relevance to any researcher doing SVD-based pruning with LLMs. All code to generate the figures is open source and available under [2].

## 2   Related Work

RMT has been widely used as a calculational tool for performing statistical averages in the analysis of machine learning models. Early applications of RMT to neural networks, such as [35], analyzed the spectral properties of loss surfaces in deep learning, providing insights into learning dynamics. Building on this foundation, Baskerville et al. [6] derived universal aspects of outliers in loss surfaces. Beyond its role in statistical analysis, RMT has been proposed as a tool for analyzing trained network weight matrices [34]. In [31], RMT was applied to weight matrices by examining the learning dynamics of image recognition models through their spectra. Following up on this work, Martin et al. [32] suggested that large outliers in the singular value spectrum are indicative of well-trained matrices. Further studies [28, 44] reinforced RMT's utility in understanding how networks evolve during training. They demonstrated that deviations from RMT predictions indicate where feature learning occurs, as opposed to *lazy learning* [12], where weights remain close to their initial random state. These findings underscore RMT's potential for identifying regions of learned features without the need for training data.

Transformers present unique challenges in understanding information storage. Prior work [25, 38] has shown that different matrix types specialize in storing distinct types of knowledge, while Aken et al. [1] examined how semantic information is encoded in neuron activations. In [39], LLMs are compressed by a low-rank approximation of the weights that optimizes for a minimal change in the activations. [49] compress LLMs based on a similar metric but transform the data on which the activations are computed first, as they find that the rank of the singular values does not represent their relevance otherwise. To overcome such issues with SVD-based approximations, several works [11, 16, 47, 50] focused on quantizing weights instead of creating low-rank approximations.

The low-rank structure of features in neural networks has been explored in [20]. Yu and Wu [52] highlight that even though the feature matrices of transformers are often low rank, their weight matrices are not, revealing a complex relationship between representations and parameters. Positional encodings, crucial to transformer performance, have also been studied for their role in shaping the learned feature space [46].

## 3   Notation and Spectral Properties of LLMs

Large language models (LLMs) are typically composed of three main parts: an initial embedding layer, a repeated stack of transformer blocks, and a final output layer. Each transformer block

---

[1] google-bert/bert-base-uncased

[2] EleutherAI/pythia-410m-deduped

[3] meta-llama/Meta-Llama-3.1-8B

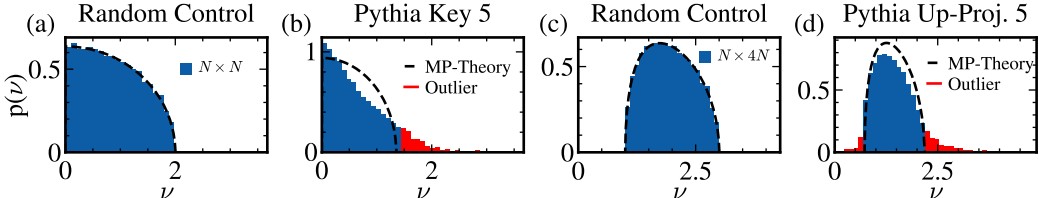

Figure 1: Spectra of random matrices (a,c) and trained matrices (b,d) in comparison to the theoretically predicted Marchenko curve for square matrices (a-b) and non-square matrices (c-d). Panel (a) shows that the theoretical prediction for a square matrix with dimension N=768 agrees perfectly with the empirical spectrum of a untrained weight matrix. Panel (b) shows that training led to the emergence of outliers in the Key matrix of the fifth block from a Pythia model. Panel (c) shows the perfect agreement of theory and initialized weight matrices in the case of non-square matrices. Square and non-square matrices differ fundamentally, as non-square matrices can also exhibit outliers to the left, as displayed in panel (d) for the Up-Projection matrix of block 5 for a Pythia model.

$B^{(l)}$, where $l \in [1, N]$ denotes the block index, has the same internal structure and weight matrix shapes. In this work, we use a consistent naming convention for the weight matrices within each transformer block. The attention sub-layer uses the Query matrix $W_Q^{(l)}$, the Key matrix $W_K^{(l)}$, the Value matrix $W_V^{(l)}$, and the Attention-Output matrix $W_O^{(l)}$. These matrices form the multi-head attention mechanism, which can be written as

$$\text{Att}(X) = \left[ \text{softmax}\left( \frac{(W_Q X)(W_K X)^\top}{\sqrt{d_k}} \right) (W_V X) \right] W_O . \tag{1}$$

Following the attention sub-layer, the feedforward sub-layer typically employs an Up-Projection matrix $W_U^{(l)}$ and a Down-Projection matrix $W_D^{(l)}$. Depending on the architecture, there may also be a Gate-Projection matrix $W_G^{(l)}$. In the case of a Gated Linear Unit (GLU), these matrices are related by

$$\text{GLU}(X) = \left[ \sigma(W_U X + b_U) \odot (W_G X + b_G) \right] W_D , \tag{2}$$

where $\odot$ denotes the Hadamard product. If no GLU is used, $W_U$ and $W_D$ are combined with a nonlinearity to form a standard feedforward layer.

To analyze these weight matrices, we use a singular value decomposition (SVD) to factor each weight matrix into its singular values and singular vectors. For a matrix $W \in \mathbb{R}^{m \times n}$, the SVD is

$$W = USV^\top , \tag{3}$$

where $U \in \mathbb{R}^{m \times m}$ and $V \in \mathbb{R}^{n \times n}$ are orthogonal matrices containing the left and right singular vectors, respectively, and $S \in \mathbb{R}^{m \times n}$ is a diagonal matrix holding the non-negative singular values.

When $q = n/m$ stays constant, $m, n \to \infty$ and the entries of $W$ are i.i.d. with finite variance $\sigma$ and zero mean, the distribution of its singular values follows the Marchenko-Pastur (MP) law [30]

$$P(\nu) = \begin{cases} \frac{q}{\pi \tilde{\sigma}^2 \nu} \sqrt{(\nu_+^2 - \nu^2)(\nu^2 - \nu_-^2)} & \nu \in [\nu_-, \nu_+] \\ 0 & \text{else} \end{cases} \tag{4}$$

$$\nu_\pm = \tilde{\sigma}(1 \pm \sqrt{1/q}) , \quad \tilde{\sigma} = \sigma \sqrt{n} , \tag{5}$$

where $n \geq m$ without loss of generality. In large, randomly initialized neural networks, the singular values of weight matrices typically approximate this distribution. After training, deviations from the MP law indicate learned structure, often manifested as outliers beyond the MP bounds $[\nu_-, \nu_+]$. Past work suggests that such outliers can signal effective feature learning [32, 44].

When plotting the theoretical Marchenko-Pastur curve of an empirical matrix, we estimate the standard deviation of the underlying matrix as described in [4]. This is used in Figure 1 where we illustrate the Marchenko-Pastur distribution together with the spectra of trained and random weight matrices. We see that for random weight matrices of BERT size, where the smaller dimension has $N = 768$, the theoretical prediction agrees perfectly with the empirical spectrum as shown in

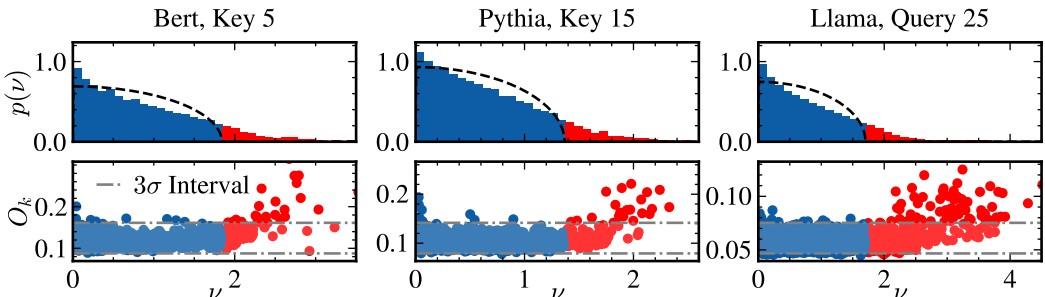

Figure 2: Spectra and maximum overlap $O_k$ of the corresponding right singular vectors with eigenvectors of the activation covariance matrix (see Eq. (7)) for square matrices of different LLMs. We observe that outliers of the Marchenko-Pastur bulk have a significantly increased overlap, indicating that important information might be carried. In the case of square matrices, singular values can only exit the bulk in the region of large singular values.

panels (a) and (c). When training weight matrices, outliers emerge as shown in panels (b) and (d) for the Key and Up-Projection matrix of Pythia. An important observation is that in the case of square matrices, outliers can only emerge towards larger values. In the case of non-square matrices, the theoretical spectrum has a minimal value that is strictly larger than zero, allowing for outliers towards small singular values. Neglecting these values to construct a low rank matrix $W'$ from $W$ is often considered optimal as the norm $|W' - W|_2$ is altered the least. However, we will show in the following that these singular values contribute significantly more to the performance of the network than larger singular values that remain in the bulk.

## 4 Overlap of Features and Weights

The following chapter shows how the singular vectors that correspond to outliers of the spectrum–small and large– can be related to important directions of the data matrix. This is done by computing the covariance matrix of the activations that enter each matrix in each block and comparing the eigenvectors of this covariance matrix to the singular vectors of the corresponding weight matrix.

Formally we define the activation covariance matrix $C(W^{(\ell)})$ of matrix $W^{(\ell)}$ using the input activations $\boldsymbol{h}(W^{(\ell)})$ to matrix $W^{(\ell)}$ and the average input activation entering that matrix $\bar{\boldsymbol{h}}(W^{(\ell)})$. Using the index $i$ as a token index that runs over the dataset, we compute

$$C = \left\langle \left(\boldsymbol{h}_i - \bar{\boldsymbol{h}}_i\right) \left(\boldsymbol{h}_i - \bar{\boldsymbol{h}}_i\right)^{\top} \right\rangle_i , \tag{6}$$

where we dropped the explicit dependence on $W^{(\ell)}$ for cleaner notation. The resulting matrix $C$ is symmetric, and therefore has an orthonormal set of eigenvectors $\boldsymbol{f}_i$ with eigenvalues $\lambda_i$.

Since the transformation of matrix $W$ is given as $W\boldsymbol{h} + \boldsymbol{b}$, where $W$ has singular value decomposition $W = U S V^{\top}$, the activations $\boldsymbol{h}$ are mapped onto the space spanned by the right singular vectors $V$. To see whether a particular eigenvector $\boldsymbol{f}_j$ of the activation covariance matrix aligns with one of the right singular vectors $\boldsymbol{v}_k$ of $W$, we compute

$$O_k = \max_j(\boldsymbol{v}_k \cdot \boldsymbol{f}_j), \quad j \in \{1, 2, ..., n\} . \tag{7}$$

This measure quantifies the extent to which a singular vector captures a specific direction of the data represented by the activation covariance matrix. In the following, we estimate $C$ from the WikiText [33] dataset. The appendix shows additional results from the BookCorpus [53] datasets.

Figure 2 shows the overlap computed based on Eq. (7) in comparison to the spectrum of our three transformers. As the displayed matrices of Bert, Pythia, and Llama are square, we find outliers from the Marchenko-Pastur distribution only on the right side of the spectrum (upper panels). The bottom panels show the overlap $O_k$ of each singular vector at the position where the corresponding singular value is in the spectrum. We observe that singular vectors corresponding to outliers of the spectrum have a strong correspondence to one of the eigenvectors of the activation covariance matrix. We can

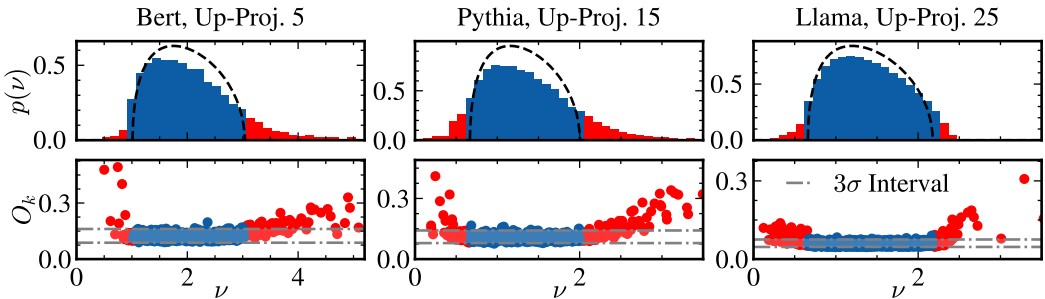

Figure 3: Spectra and maximum overlap $O_k$ of the corresponding right singular vectors with eigenvectors of the activation covariance matrix for rectangular matrices of different LLMs. The singular vectors corresponding to large singular values exhibit a significant overlap with the activation covariance matrix. However, in the case of non-square matrices, we also observe an increased overlap with the singular vectors corresponding to the smallest singular values. This is surprising as the smallest singular values contribute very little to the variance of the activations in the next layer and are typically regarded as something that can be discarded.

interpret these singular vectors as picking up a direction of the data with high variance and weighting it highly. For reference, the grey lines display the $3\sigma$ interval when assuming singular vectors from a random matrix. We find several statistically highly significant points above these lines.

In Figure 3 we show rectangular matrices of all three transformers, where in this case, outliers of the spectrum can emerge towards large values and small values. Interestingly, we observe that singular vectors of both groups have strong correspondence to an eigenvector, indicating that the singular vectors corresponding to the smallest singular values encode properties of the activation covariance matrix and hence of the data! This is an exceptional result as small singular values contribute the least to the overall matrix with respect to the L2 norm, and they are the first ones to be discarded when pruning naively. The results are again statistically highly significant even with respect to a $3\sigma$ interval.

The two plots illustrated the relation between small singular values, and the data for specific matrices. To further look into this phenomenon, we present all matrix types of block 10 for both Pythia and Llama in Figure 4. Other blocks are displayed in the appendix. We see that for Pythia (upper 2 panels) Query, Key, and Value matrix exhibit increased overlap only for their largest singular values. The Attention-Output matrix has no significant overlap with the activation covariance matrix, a phenomenon that holds for BERT, Pythia, and Llama, as elaborated in the appendix. The Up and Down-Projection matrices exhibit significantly increased overlaps for both their largest and smallest singular values. However, we also find singular values with small overlaps in this region. This is different for Llama's Up and Gated-Projection matrix, where the small and large singular values significantly overlap with the covariance matrix. Interestingly, the singular vectors of the Down-Projection matrix have no increased overlap with any of the eigenvectors from the activation covariance matrix. We suspect that the placement of this matrix at the end of the Gated Linear Unit—rather than after a separate nonlinearity—may render certain weights redundant, leading to no significant overlap. Surprisingly, we find that the Key and Value matrix do not exhibit large overlaps for their smallest singular values while being non-square matrices. We speculate that this arises from their indirect relation to the activations of the previous layer, which are processed in the Attention mechanism. The development of the overlap as training progresses can be found in the appendix.

## 5 Contribution of Singular Values to Model Performance

In the preceding section, we observed a strong overlap between the singular vectors of weight matrices – particularly in regions outside the Marchenko-Pastur (MP) boundary – and the eigenvectors of the activation covariance matrix. To further evaluate the importance of these singular values and their corresponding vectors, we conduct experiments in which we selectively remove certain groups of singular values. Specifically, removing a singular value $\nu_r$ from a weight matrix $W$ involves zeroing

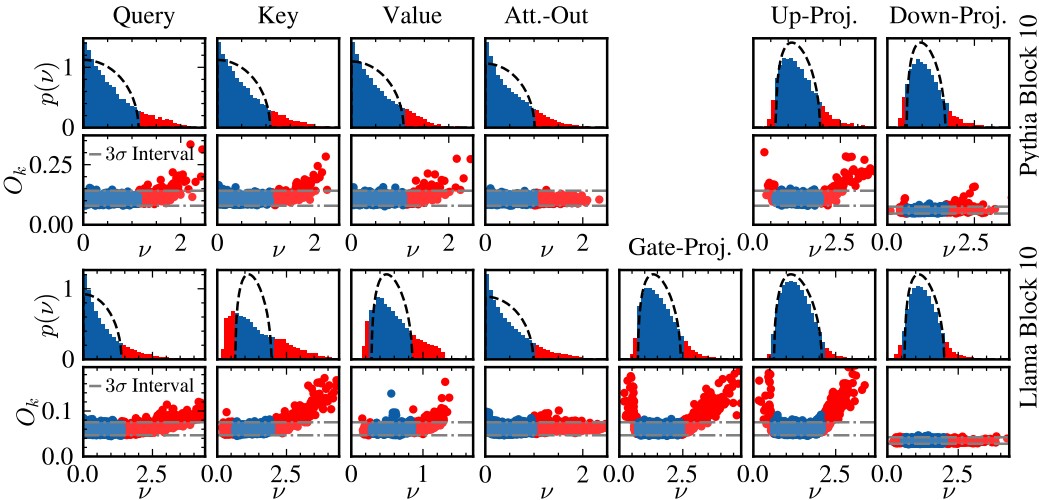

Figure 4: Spectra and the overlap $O_k$ for block 10 of Pythia and Llama. We find that in the case of non-square matrices, the singular value outliers to the left of the spectrum can have a significantly increased overlap $O_k$ with the eigenvectors of the activation covariance matrix. This is the case for the Up-Projection matrix of Pythia and Llama, as well as Llama's Gate-Projection matrix. In case of the Attention-Output matrices and the Down-Projection matrix of Llama, we find very little overlap. This is a systematic finding and may reflect the training dynamics (see Appendix for details).

out $\nu_r$ in the diagonal matrix $S$ of its singular value decomposition,

$$W = USV^\top, \tag{8}$$

$$\tilde{S}_{ii} = \begin{cases} 0 & \text{for } i = r \\ \nu_i & \text{else} \end{cases} \quad \longrightarrow \quad \tilde{W} = U\tilde{S}V^\top . \tag{9}$$

We then reconstruct the weight matrix $\tilde{W}$ using the original singular vectors. To compare the relative impact of different parts of the spectrum, we group the rank-ordered singular values of each matrix into ten equal-sized deciles, with the smallest $10\%$ in the first decile and the largest $10\%$ in the tenth. We apply this procedure by choosing one matrix type (e.g., Query) and zeroing out one of the ten deciles in all weight matrices of that type. We then measure the resulting increase in $\Delta$Perplexity $= \text{PPL}(f_{\tilde{W}}) - \text{PPL}(f_W)$, on the WikiText dataset. The results are shown in Figure 5, where removing the largest singular values consistently causes a large perplexity increase in all matrix types for both Pythia and Llama. This is expected as the largest singular values and corresponding vectors are the directions that describe the largest percentage of the variance of the next layers' activations.

When going through the deciles from large to small, we observe that the contribution of the deciles decreases monotonically for all square matrices. For the non-square matrices, the largest nine deciles also follow this trend. **However, for non-square matrices, the smallest decile always contributes more than some of the larger deciles.** This observation underscores that valuable information can be encoded at the lower end of the spectrum, potentially explaining why standard pruning strategies often struggle with transformers [3].

These findings are further backed up by Llama results on the GSM8K benchmark, which tests basic math problems. These results are depicted in Table 1, where decile one corresponds to the smallest and decile ten to the largest singular values. For the Down-Projection matrix, we find that the smallest singular values (decile 1) are the second most important decile, which is in excellent agreement with our previous results for this rectangular matrix (compare Figure 5). In contrast, considering the quadratic Attention-Output matrix, the smallest singular values are one of the least important deciles, in agreement with theoretical expectations and the previous observations. The smallest singular values of the non-quadratic Gate-Projection matrix are the fourth important decile, also in excellent agreement with the results of Figure 5. For the quadratic Query matrix, only the two largest deciles appear to be important. Further results for the HumanEval benchmark show a similar pattern and are displayed in the Appendix.

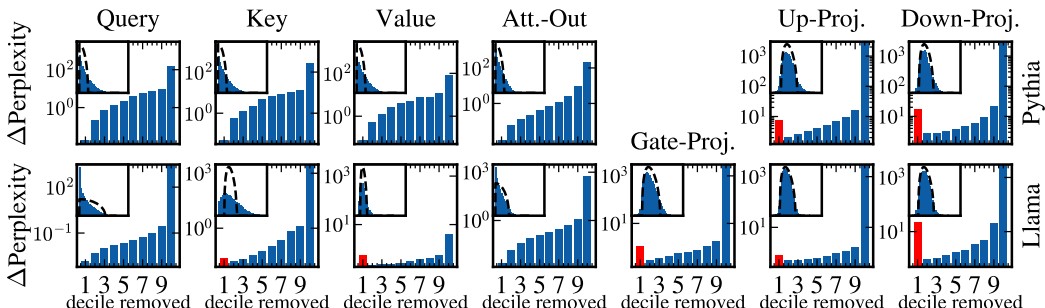

Figure 5: Increase in perplexity on WikiText for Pythia and Llama when removing deciles of rank-ordered singular values. Singular value deciles are removed from all blocks, but only from a specific matrix type, e.g., all Key matrices. The inset shows the respective spectra averaged over all blocks. As expected, removing the largest singular values substantially affects perplexity for both models and all matrix types, as the matrix changes significantly upon the removal. Interestingly, we find for non-square matrices, i.e., matrices with spectra that have outliers towards small values, that the decile with the smallest singular values is more important than some of the larger deciles. In case of the Down-Projection matrix, these singular values are even the second most important group. These findings confirm that meaningful information may reside in the lower end of the spectrum.

We provide further experiments for removing deciles from all weight matrix types for LLaMA 3.1 8B Chat on the RULER [22] benchmark for a context length of 8192. The tasks needle in a haystack (niah) were evaluated with several values to be extracted (niah multiV) and for several queries (niah multiQ). We also conducted experiments on value tracking (vt), common word extraction (cwe), and frequent words extraction (fwe). The results when removing singular values from all layers simultaneously are shown in table 2. We find that the smallest singular values are even the second most important decile in this task.

For Bert we additionally conduct fine-tuning experiments on BoolQ [13], rte [8, 19, 21] and SST2 [42]. We compare two scenarios: (i) removing a particular decile of singular values before fine-tuning and then fine-tuning the altered model, versus (ii) fine-tuning the model first and then removing the singular value decile. Now, we are removing deciles from all weight matrices simultaneously to compare to previous literature results and estimate the $3\sigma$ interval from the standard deviation of six runs without pruning. Figure 6 shows the results when removing singular value deciles from Bert, where in case (i) –left pannel, prune first–, removing small singular values reduces the accuracy slightly but statistically significantly in the case of RTE and SST22. For BoolQ, the lost knowledge was either recovered or not necessary. By contrast, in case (ii) –right panel, fine-tune first–, removing the same small singular values degrades accuracy significantly in all cases, showcasing their surprising relevance. Large singular values (decile ten) are consistently important, as removing them drops performance to near random guessing (off-scale points in the right panel). This finding provides insights into a recent debate about the importance of small singular values in transformers: some argue that they are essential for good performance [23], while others report performance gains from their removal [41]. Our results clarify this conflict by showing that small singular values do matter,

Table 1: Effect of removing deciles of singular values from various weight matrices of Llama3-8B on the GSM-8K accuracy under 3-shot prompting. Decile 1 corresponds to the smallest singular values, and Decile 10 to the largest. Removal of the smallest singular values from the non-square Down-Projection matrix significantly reduces accuracy (from the 43.2% baseline to 2%), confirming their importance. The smallest singular values of the square Attention-Output and Query matrices are not particularly relevant.

| Matrix | Dec. 1 | Dec. 2 | Dec. 3 | Dec. 4 | Dec. 5 | Dec. 6 | Dec. 7 | Dec. 8 | Dec. 9 | Dec. 10 |
|---|---|---|---|---|---|---|---|---|---|---|
| Down-Proj. | 2.0% | 30.2% | 28.0% | 27.1% | 26.2% | 22.0% | 11.0% | 15.0% | 5.0% | 0.0% |
| Att.-Out. | 40.0% | 41.8% | 39.0% | 38.8% | 39.8% | 37.1% | 37.6% | 31.4% | 25.1% | 0.0% |
| Gate-Proj. | 34.1% | 37.0% | 39.1% | 39.6% | 38.1% | 38.0% | 35.9% | 33.3% | 23.6% | 0.0% |
| Query | 40.3% | 44.3% | 41.4% | 40.5% | 39.5% | 40.1% | 40.7% | 40.7% | 35.8% | 0.0% |

Table 2: LLaMA 3.1 8B Chat results on RULER benchmark tasks (context length 8192) when removing singular value deciles from all layers simultaneously.

| Task | Dec.1 | Dec.2 | Dec.3 | Dec.4 | Dec.5 | Dec.6 | Dec.7 | Dec.8 | Dec.9 | Dec.10 |
|---|---|---|---|---|---|---|---|---|---|---|
| niah multiV | 0.0 | 61.0 | 88.5 | 7.0 | 65.5 | 99.25 | 13.25 | 0.0 | 0.0 | 0.0 |
| niah multiQ | 0.0 | 69.5 | 80.5 | 0.0 | 57.75 | 99.0 | 11.75 | 0.0 | 0.0 | 0.0 |
| vt | 0.0 | 60.6 | 56.2 | 33.6 | 6.4 | 47.6 | 0.0 | 0.0 | 0.0 | 0.0 |
| cwe | 0.0 | 57.8 | 37.8 | 21.0 | 13.7 | 8.3 | 9.0 | 3.2 | 0.0 | 0.0 |
| fwe | 0.0 | 86.33 | 83.0 | 69.33 | 17.33 | 84.0 | 33.67 | 3.0 | 2.0 | 0.0 |

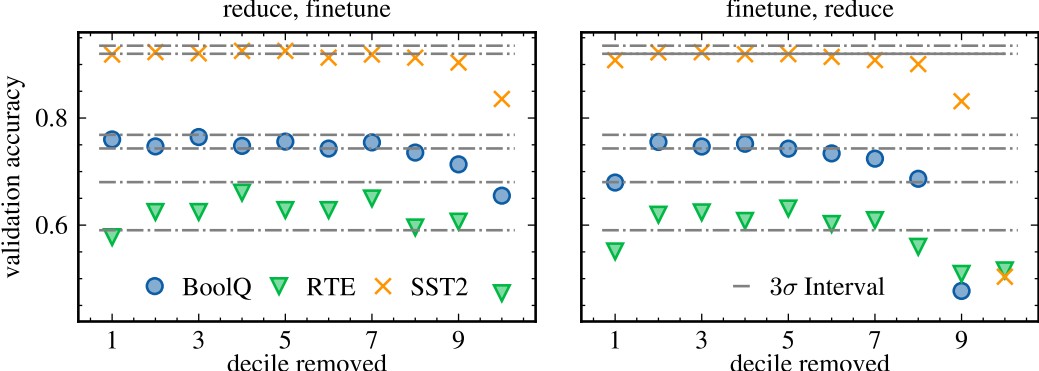

Figure 6: Effect on validation accuracy on BoolQ, RTE, and SST2 when removing deciles of singular values from all matrices (except embedding weights) in a Bert transformer. Decile ten corresponds to the largest 10% of singular values, and its removal (both before and after fine-tuning) significantly lowers accuracy (left and right panels). For the smallest singular values (decile one), the two scenarios differ. When the smallest values are removed *before* fine-tuning, final accuracy is either fully recovered, as in the case of BoolQ, or slightly reduced (left panel). If the model is fine-tuned *first* and then has its smallest singular values removed, accuracy declines statistically significantly for all three datasets (right). This indicates that the information that is stored in small singular values, along with their corresponding vectors, can sometimes be recovered by fine-tuning or is unused in a specific dataset, as seen in the case of BoolQ.

but mainly once the model has been fine-tuned. Hsu et al. [23], fine-tune before pruning, and observed that small singular values are critical, while Sharma et al. [41] found gains from pruning them, and evaluating the model *without* additional fine-tuning. We interpret this behavior as evidence that fine-tuning – and potentially alignment– relies on small singular values and their associated vectors. Since alignment may degrade performance on reasoning tasks [36], removing these directions can improve task metrics. However, doing so after alignment could unintentionally erase that alignment.

## 6 Minimal Random Matrix Theory Model

The aim of this section is to provide a minimal RMT model for the occurrence of small singular value outliers. For this, we consider a simple two-layer linear student model, trained on Gaussian inputs, and labels produced by a linear rank-1 teacher. The effects of non-trivial correlations in the noise depend on whether the noise has predominantly small or large covariance in the direction of the outlier.

A common model to describe weight matrices $W$ consists of a low-rank matrix $W_0$ encoding the rule and a noise matrix $X$ as $W = W_0 + X$. Under the common assumption [24, 43] of white noise, $\langle X \rangle = 0$, $\mathrm{Cov}(X) = \mathbb{1}$, singular values of $W_0$ can only be outliers in $W$ if they lie above the Marchenko-Pastur bulk of $X$, i.e. above the BBP critical value [5, 7, 10]. In the following, we present a model with a non-trivial covariance of the noise, which can have a singular value of W below the MP bulk.

We start from a teacher-student setup with teacher function $\sigma_T(\boldsymbol{\xi}) = \boldsymbol{u}^\top \lambda \boldsymbol{\xi}$ with normalized $\boldsymbol{u} \in \mathbb{R}^{N \times 1}$ and $\lambda \in \mathbb{R}$. The student is a linear two-layer network $\sigma_S(\boldsymbol{\xi}) = \boldsymbol{v}^\top W \boldsymbol{\xi}$ with normalized, fixed second-layer weight $\boldsymbol{v} \in \mathbb{R}^{K \times 1}$ and trained weight matrix $W \in \mathbb{R}^{K \times N}$ of aspect ratio $q = K/N < 1$. Here, $\boldsymbol{v}$ may be interpreted as a learned, stabilized weight singular vector in a subsequent layer of a real network. We show in the following that this model explains the occurrence of small outliers below the MP bulk, and that it relates the amount of information contained in the outlier singular vector to the distance of the outlier to the bulk.

We consider a mean-squared-error loss function $\mathcal{L}(\boldsymbol{\xi}) = (\boldsymbol{u}^\top \lambda \boldsymbol{\xi} - \boldsymbol{v}^\top W \boldsymbol{\xi})^2 / 2$ for inputs $\boldsymbol{\xi} \in \mathcal{N}(0, 1)$, giving rise to a generalization error

$$\epsilon_g(W) := \langle \mathcal{L}(\xi) \rangle_\xi = \frac{1}{2} \mathrm{Tr} \left[ (\boldsymbol{u}^\top \lambda - \boldsymbol{v}^\top W)^2 \right] . \tag{10}$$

Here we adopted the notation $A^2 = A^\top A$. We study the weight matrix ensemble

$$P(W) \propto \exp \left[ -N\beta\epsilon_g(W) - \frac{N}{2\alpha} \mathrm{Tr}(W^\top W) \right] \propto \exp \left[ -\frac{N}{2} \mathrm{Tr} \left[ (W - W_0)^\top \Sigma^{-1} (W - W_0) \right] \right] , \tag{11}$$

where $\beta\alpha \to \infty$ corresponds to perfect learning, and the $\alpha$ term has the role of a prior for weight matrices at initialization or an $L_2$ regularization strength. In an annealed approximation, $\beta$ corresponds to the number of examples shown to the network divided by the input dimension [17]. Here, the mean is $W_0 = \frac{\alpha\beta\lambda}{1+\alpha\beta} \boldsymbol{v}\boldsymbol{u}^\top$, and the correlation matrix is given by

$$\Sigma = \alpha \mathbb{1} - \frac{\alpha^2\beta}{1 + \alpha\beta} \boldsymbol{v}\boldsymbol{v}^\top . \tag{12}$$

Weight matrices of this matrix ensemble have the form $W = W_0 + X$, with a random matrix $X$ of zero mean and row covariance matrix $\Sigma$ (while columns are uncorrelated). For $0 < (1 + \alpha\beta)^{-1} < 1$, the noise matrix $X$ has an outlier singular value below its Marchenko-Pastur bulk. This stabilizes a small singular value outlier of $W$ if $\lambda < \sqrt{\alpha(1-q)}$ at [18]

$$\langle \nu_{\min} \rangle = \left[ \alpha \frac{1 + \alpha\beta(1 + \beta\lambda^2)}{(1 + \alpha\beta)^2} + \frac{q\alpha}{1 - \frac{(1+\alpha\beta)^2}{1+\alpha\beta(1+\beta\lambda^2)}} \right]^{1/2} \tag{13}$$

For the case of perfect learning, i.e. $\beta \to \infty$, the outlier singular value approaches $\lambda\sqrt{1 - \frac{\alpha q}{\alpha - \lambda^2}}$, only differing from $\lambda$ of $W_0$ by the level repulsion with the bulk.

In Fig. 7, we depict $\langle \nu_{\min} \rangle$ (left panel) and the singular vector overlaps (right panel) $\langle |\tilde{\boldsymbol{u}} \cdot \boldsymbol{u}| \rangle$ (blue) and $\langle |\tilde{\boldsymbol{v}} \cdot \boldsymbol{v}| \rangle$ (red) of the corresponding singular vectors $\tilde{\boldsymbol{u}}, \tilde{\boldsymbol{v}}$ of $W = \tilde{V}\tilde{S}\tilde{U}^\top$ as a function of $1/(1 + \alpha\beta)$. We find that ideal learning, $1/(1 + \alpha\beta) \to 0$, corresponds to an outlier maximally separated from the MP bulk with best overlap of the corresponding singular vectors of $W$ with those of $W_0$. For poorer learning, the singular value $\nu_{\min}$ moves towards the bulk and the vector overlap $\langle |\tilde{\boldsymbol{u}} \cdot \boldsymbol{u}| \rangle$ approaches the expected overlap between random vectors (blue, dashed line), when outlier and MP bulk fuse at $\beta_{\text{bulk}} = \alpha^2 \left( 1 - \sqrt{\alpha^2 - 4\sqrt{q}\alpha\lambda^2} \right) / 2\lambda^2$.

**Limitations**

We show empirical evidence that the singular vectors corresponding to the smallest singular values encode important directions in the data. In case of the Down-Projection matrix of Llama and the Attention-Output matrices we do not observe an enhanced overlap with the activation covariance matrix, while the smallest singular values of the Down-Projection matrix are still important for model performance, as the perplexity increases upon their removal. We argue that this is due to the lack of a non-linearity in these layers, however further research is needed to clarify this. Similarly, we argue that the laziness of the Attention-Output matrix is responsible for the lack of overlap with the activations, which needs to be verified in more detail. To further strengthen the validity of our theoretical model, the covariance of the noise present in multiple LLM weight matrices should be computed, which is only possible when doing multiple training runs.

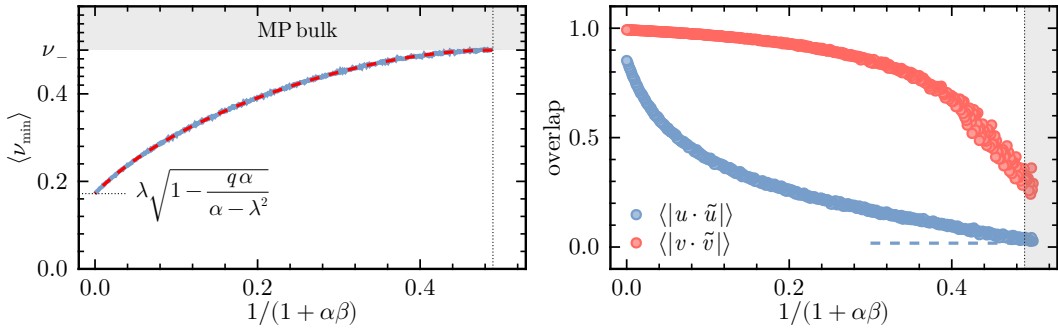

Figure 7: Left panel: Smallest singular value $\langle \nu_{\min} \rangle$ as a function of learning parameters $\alpha$ and $\beta$ where $1/(1 + \alpha\beta) = 0$ corresponds to perfect learning. We observe that $\langle \nu_{\min} \rangle$ lies outside the Marchenko-Pastur bulk in the case of good learning (small values of $1/(1 + \alpha\beta)$). The red line corresponds to our analytical result Eq (13), while the blue line corresponds to empirical values determined by drawing 100 matrices from the ensemble. Right panel: Overlap of the learned vectors $\tilde{v}$ and $\tilde{u}$ with the corresponding teacher vectors in $W_0$ as a function of the learning parameters. The results are averaged over 100 data points. The blue, dashed line depicts the expected overlaps of random vectors of length $N$. For perfect learning with $1/(1 + \beta\alpha) = 0$, the overlap of the singular vectors with those of $W_0$ is maximized, while there is a strong decrease in the overlap as $\langle \nu_{\min} \rangle$ approaches the bulk. The results in both panels are computed for $\alpha = 1$, $\lambda = 0.2$, $N = 2048$, and $K = 512$.

## 7 Conclusion

In this paper, we analyzed the singular value spectra and activation covariance matrices of Bert, Pythia, and Llama-8B models. We demonstrated that the spectra deviate from the Marchenko-Pastur distribution, not only for large singular values, but also for small singular values in the case of non-square matrices. The singular vectors corresponding to the singular values outside of the theoretically predicted spectrum were shown to be related to specific directions in the activations entering that matrix. When removing these singular values in deciles for Llama and Pythia, we found that while the largest singular values are the most important decile in all cases, the smallest singular values can even be the second most important decile. This importance is established for non-square matrices, where the smallest singular values fall outside of the Marchenko-Pastur region.

Furthermore, our fine-tuning results shed light on the debate over the relevance of small singular values in LLMs. While removing small singular values after fine-tuning drastically lowers model performance, removing them beforehand sometimes has no statistical effect on the final task, explaining some discrepancies in the literature. Using a random-matrix model we demonstrate that for singular values well below the MP-bulk, the associated singular vectors are more similar to the teacher vector than for singular values close to or inside the bulk. Thus, small singular values and associated vectors carry more information than larger ones.

In summary, this work demonstrates the importance of small singular values, showcasing when and where they are important, while additionally providing a theoretical basis for how this phenomenon can occur. We expect these insights to be crucial for future SVD-based pruning algorithms.

## Acknowledgments and Disclosure of Funding

Computations for this work were done using resources of the Leipzig University Computing Center.

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

Figure 8: Block 5 for Pythia and Llama showcasing the overlap between right singular vectors and eigenvectors of the activation covariance matrix computed on the WikiText dataset. We observe that the singular vectors corresponding to singular values outside the Marchenko-Pastur region have a significantly increased overlap with the eigenvectors of the covariance matrix as computed in Equation (7).

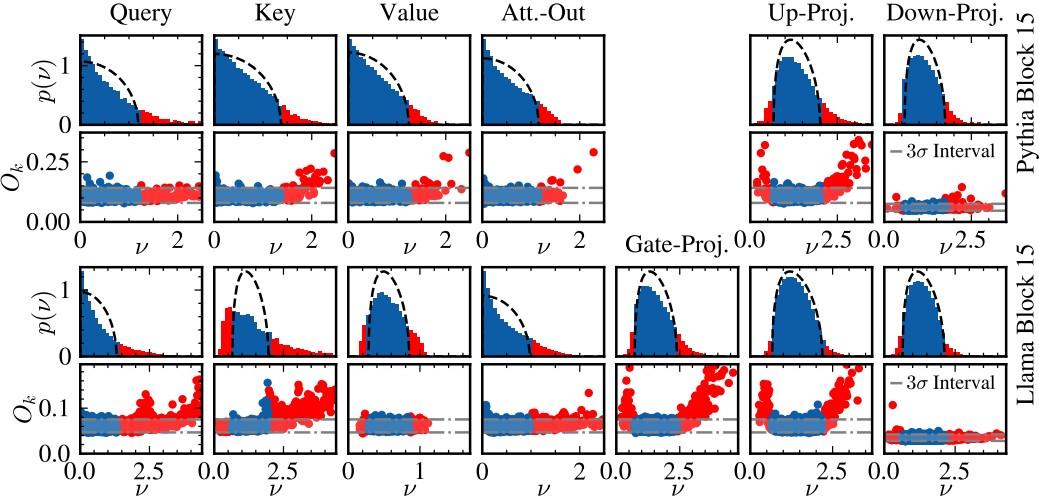

Figure 9: Block 15 for Pythia and Llama showcasing the overlap between right singular vectors and eigenvectors of the activation covariance matrix computed on the WikiText dataset. We observe that the singular vectors corresponding to singular values outside the Marchenko-Pastur region have a significantly increased overlap with the eigenvectors of the covariance matrix as computed in Equation (7).

## A  Activation Covariance Overlap

We show additional results that demonstrate the generality of the observed overlaps between the right singular vectors of the weight matrix and the eigenvectors of the corresponding activation covariance matrix. The results for block 5 of Llama and Pythia are provided in Fig. 8, while the results for block 15 are shown in Fig. 9. When singular values deviate from the Marchenko-Pastur distribution, we again find a significant overlap of the corresponding singular vectors and the eigenvectors of the activation covariance matrix, particularly in the case of small singular values.

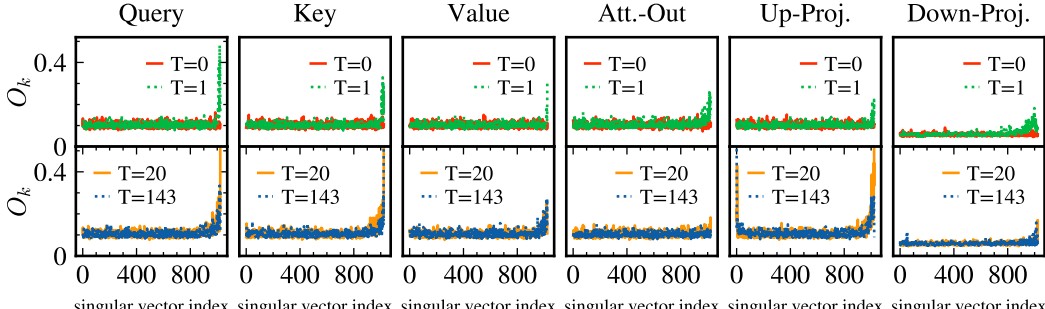

Figure 10: Time evolution of the overlap between the activation covariance matrix and the weight matrices during pre-training of Pythia (block 10), computed on WikiText. Here, we plot the overlap by singular vector index with the one corresponding to the smallest singular value on the left. At $T = 0$, matrices and activations are random, giving no overlap. By $T = 1$ (1000 gradient updates), large singular-value directions already begin to align with the activation covariance for every weight matrix. However, Attention-Output loses this overlap at later stages. Notably, the smallest singular values (e.g., in the Down-Projection matrix) only gain significant overlap in the later phases of pretraining.

To further study the outliers, in particular the ones corresponding to smaller singular values, we additionally analyze the temporal evolution. Fig. 10 shows the overlap of the activation covariance matrix and the weight matrix in the training of Pythia for block 10, computed on the WikiText dataset. As expected, there is no overlap at initialization as both the weight and the activation covariance matrix are random. Interestingly, already at $T = 1$ (corresponding to 1000 gradient updates), the model forms a significant overlap with the activation covariance matrix for all matrix types. In case of the Attention-Output matrix, this overlap is later reduced, while the overlap of Query, Key, and Value increases significantly, which we interpret as repositioning of the information in the weight matrices. The overlap of the singular vectors corresponding to small singular values emerges in the later stages of training for the Down-Projection matrix and increases drastically for the Up-Projection matrix. Considering that fine-tuning can significantly impact the smallest singular values, we speculate that the smallest singular values can be related to potentially more sophisticated concepts, which may be learned in the later stages of pre-training.

## B    Additional Results on Book Corpus

To show that the results presented in the manuscript are of a general form and do not depend on the WikiText dataset, we conduct additional experiments on the Book Corpus dataset. Figure 11 confirms that for all non-square matrices, the smallest singular values are more important than some of the larger deciles. In case of the Llama Down-Projection matrix, the smallest decile is even the second most important group. These results can be attributed to the same phenomenon as shown in Figure 12. The overlap between singular vectors and eigenvectors of the activation covariance is very similar to the one observed in the main manuscript, showcasing large overlaps for some of the smaller singular values.

## C    Special Feature of the Activation Covariance Basis

When computing the overlaps between a particular singular vector and the eigenvectors of the activation covariance basis, we defined the quantity

$$O_k = \max_j(\boldsymbol{v}_k \cdot \boldsymbol{f}_j), \quad j \in \{1, 2, ..., n\} . \tag{14}$$

as the maximum coefficient when developing the singular vector $\boldsymbol{v}$ in the basis of the eigenvectors $\boldsymbol{f}_j$. If one of the 768-4096 coefficients is large, we argue that these two vectors have a strong correspondence. A natural questions is the behavior of the quantity in a different basis. In principal there exist a trivial basis for which the overlap is exactly one, i.e. the singular vectors $\boldsymbol{v}_j$ themselves.

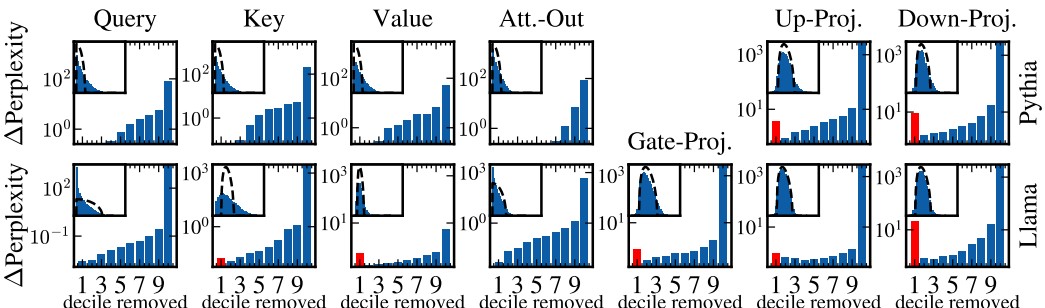

Figure 11: Increase in perplexity on the Book Corpus dataset for Pythia and Llama when removing deciles of rank-ordered singular values. Singular value deciles are removed from all blocks, but only from a specific matrix type, e.g., all Key matrices. The inset shows the respective spectra averaged over all blocks. Removing the largest singular values substantially affects perplexity for both models and all matrix types, as the matrices change significantly. As in the case of WikiText, we find that for non-square matrices, the decile with the smallest singular values is more important than some of the larger deciles.

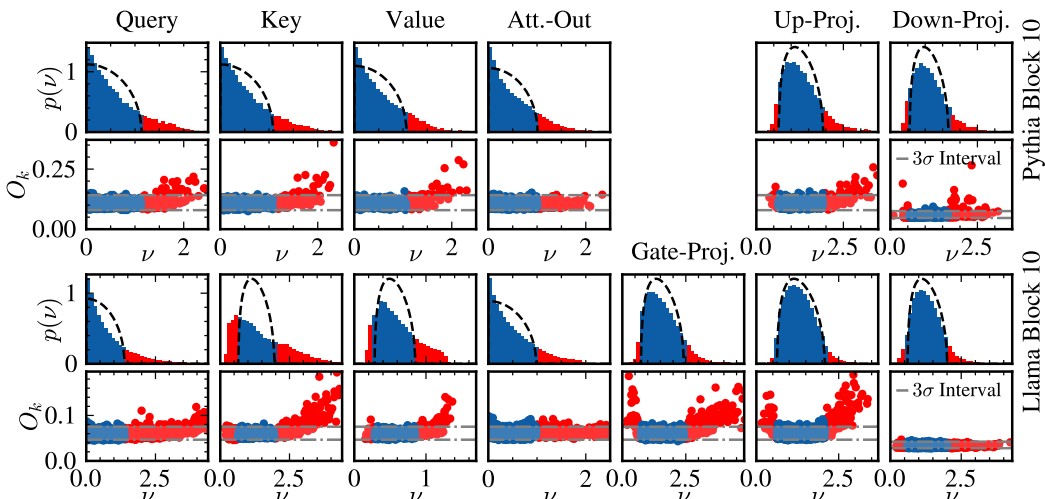

Figure 12: Block 10 for Pythia and Llama, showcasing the overlap between right singular vectors and eigenvectors of the activation covariance matrix when the activation covariance matrix is computed based on the Book Corpus dataset. We observe that the singular vectors corresponding to singular values outside the Marchenko-Pastur region have a significantly increased overlap with the eigenvectors of the covariance matrix.

In Figure 13 we display the quantity

$$M_k = \max_j(\boldsymbol{v}_k \cdot \boldsymbol{e}_j), \quad j \in \{1, 2, ..., n\} . \tag{15}$$

which is just the maximal component of each singular vector. For Pythia, $M_k$ is small for the singular vectors corresponding to largest singular values in all cases but the Down-Projection matrix, showcasing that the activation covariance basis is a special basis in this case. For Llama we observe a different picture where the new basis has almost a complementary behavior to the activation covariance eigenbasis. While the Attention-Output and Down-Projection matrix of Llama do not show increased overlaps $O_k$, they appear to be strongly localized, indicated by large $M_k$. This is an interesting additional observation and may explain the large perplexity increases when removing the smallest singular values of the Down-Projection matrix.

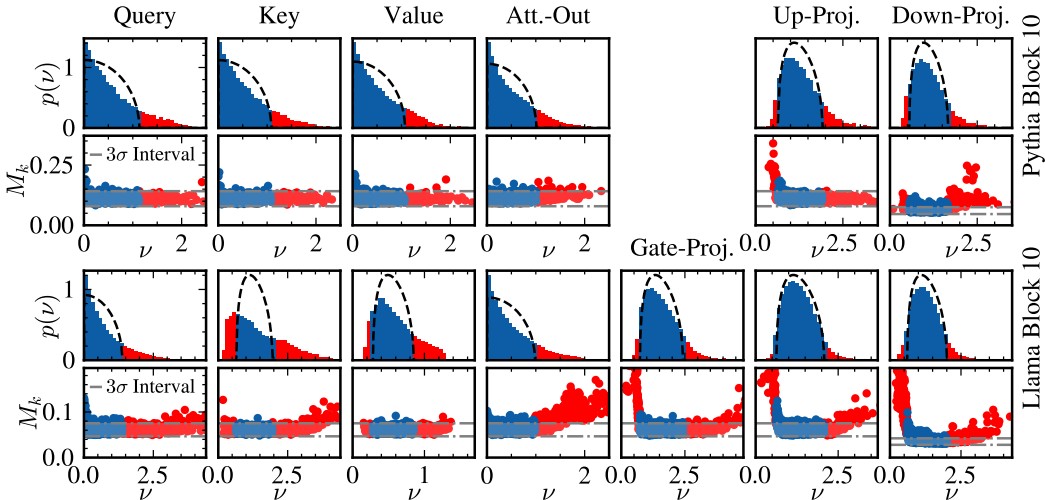

Figure 13: Block 10 for Pythia and Llama, showcasing the overlap between right singular vectors in the basis of standard unit vectors $e_i$. We observe that this basis does not have a large maximum overlap with the singular vectors for largest singular values in most cases. However, it appears to be complementary to the activation covariance matrix in the case of the Down-Projection matrix and Attention-Output matrix of Llama where we observe strong overlaps.

## D  Lazy Learning Regime

It is possible to train neural networks in the lazy regime where the final weights of the trained model are very close to the initial ones [12]. By rescaling the input of the softmax function in the final layer by a constant $\alpha > 1$

$$a_L = \text{softmax}\left(\alpha(\mathsf{W}_L \boldsymbol{a}_{L-1} + \boldsymbol{b}_L)\right) \ ,$$

we achieve that very small changes in the output logits prior to the softmax function have a large effect on the output after the softmax function. To allow for learning with a usual learning rate, the loss is changed to

$$l(\boldsymbol{W}, \boldsymbol{b}) = -\frac{1}{N\alpha^2} \sum_{k=1}^{N} \boldsymbol{y}^{(k)} \cdot \ln(\boldsymbol{a}_{\text{out}}^{(k)}) \ , \tag{16}$$

to incorporate the large differences in the output activations $a_L$ induced by small weight changes. When training such networks, previous studies [28, 44] demonstrated that RMT properties stay intact in the case of lazy learning, where weights remain close to their initial random state. However, large outliers (i.e. singular values outside of the MP-boundary $[\nu_-, \nu_+]$) often reflect critical learned features leading to the results that models trained in the lazy regime generally perform worse than models trained in the rich or feature learning regime. In Figure 14 we compare the average singular value spectrum of all blocks to its initial state for both the Query and Attention-Output matrix. We find that for all three models, the Attention-Output matrix has significantly fewer outliers. Considering that there is no overlap with the activation covariance matrix either, we speculate that the Attention-Output matrix may be trained in the lazy regime.

## E  Computational Resources

The computational resources needed to compute the shown results are mostly defined from the perplexity computations. Computing spectra and the activation covariance matrices for all models can be estimated as less than 100 GPU hours on a V100. To compute the perplexity on subparts of the WikiText and BookCorpus dataset several times with different parts of the model removed, we estimate less than 1000 GPU hours.

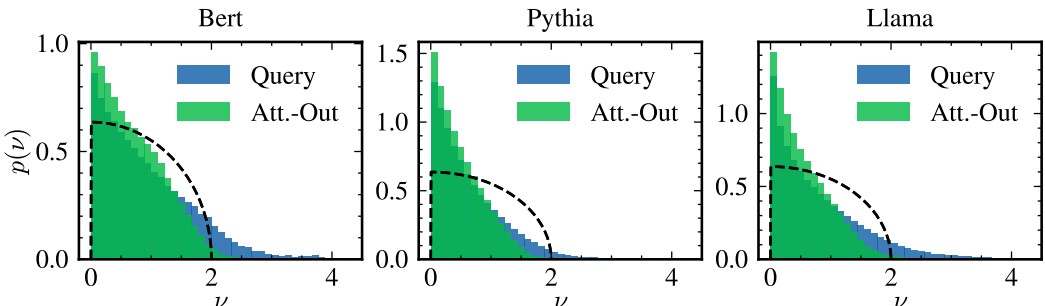

Figure 14: Averaged Spectra of Bert, Pythia, and Llama compared to the initial distribution of singular values for Query and Attention-Output matrices. We observe that the Attention-Output matrix develops substantially fewer outliers than the Query matrices, which we interpret as a sign that the Attention-Output matrix may be trained in the lazy regime.

## F   Details on Random Matrix Model

As described in the main text, we consider the matrix ensemble

$$P(W) = \frac{1}{\mathcal{Z}(\alpha, \beta, \lambda)} \exp\left[-N\beta\epsilon_g(W) - \frac{N}{2\alpha}\text{Tr}(W^\top W)\right] . \tag{17}$$

Using the expression for the generalization error Eq. (10), we first complete the square

$$2\beta\epsilon_g(W) + \frac{1}{\alpha}\text{Tr}(W^\top W) = \text{Tr}\left[(W - W_0)^\top \Sigma^{-1}(W - W_0)\right] \tag{18}$$

with

$$W_0 = \frac{\alpha\beta\lambda}{1 + \alpha\beta} \boldsymbol{v}\boldsymbol{u}^\top \tag{19}$$

$$\Sigma^{-1} = \frac{1}{\alpha}\mathbb{1} + \beta\boldsymbol{v}\boldsymbol{v}^\top . \tag{20}$$

Computing the inverse of $\Sigma^{-1}$ yields Eq. (12).

We compute the partition function, i.e. the normalization, by performing the Gaussian integral

$$\mathcal{Z}(\alpha, \beta, \lambda) = \int dW \exp\left[-\frac{N}{2}(W - W_0)^\top \Sigma^{-1}(W - W_0)\right] \tag{21}$$

$$= (2\pi)^{\frac{NK}{2}} \exp\left[-\frac{\lambda^2}{2}\frac{N\beta}{1 + \alpha\beta}\right]\left(\frac{N}{\alpha}\right)^{-\frac{NK}{2}}(1 + \beta\alpha)^{-\frac{N}{2}} . \tag{22}$$

We can then compute the generalization error

$$\langle\epsilon_g(W)\rangle = -\frac{1}{N}\frac{\partial\mathcal{Z}(\alpha, \beta, \lambda)}{\partial\beta} = \frac{\alpha}{2(1 + \beta\alpha)} + \frac{\lambda^2}{2(1 + \beta\alpha)^2} . \tag{23}$$

The expectation value of the generalization error is minimal for largest separation ($1/(1 + \alpha\beta) \to 0$) of the observed singular value $\nu_{\min}$ from the MP bulk of the spectrum of $W$, suggesting that moving singular values of relevant directions out of the bulk can occur naturally when minimizing the loss during training.

In addition, the expectation value of the small outlier singular value $\langle\nu_{\min}\rangle$, Eq. (13), can be analytically obtained in the limit $N \to \infty$, $q = K/N = \text{const}$. For this, the outlier eigenvalue $\eta_{\min} \equiv \alpha - \alpha^2\beta/(1 + \alpha\beta) + (\lambda\alpha\beta/(1 + \alpha\beta))^2$ of $\Sigma + W_0^\top W_0$ is related to $\langle\nu_{\min}\rangle^2$ via the blue function [18],

$$B(y) = \frac{1}{y} + q\int_{-\infty}^{\infty}\frac{xg_\Sigma(x)}{1 - xy}\,dx \tag{24}$$

$$\langle\nu_{\min}\rangle^2 = B(1/\eta_{\min}) , \tag{25}$$

where $g_\Sigma(x) = \delta(x - \alpha)$ is the density of $\Sigma$ eigenvalues in the $K \to \infty$ limit, for which the single outlier contribution is negligible. Thus $B(y) = 1/y + q\alpha/(1 - \alpha y)$ and $\langle \nu_{\min} \rangle$ follows from Eq. (25) as given in Eq. (13).

## G   Layer Specific Statistics

As a first step towards providing more comprehensive statistics for each layer, we present outlier counts for several layers of Llama3-8B in the format [number of left outliers | number of right outliers] in Table 3. We observe that for some matrices related to the attention mechanism (Query, Key, Value, Att.-Output), the number of outliers to the right of the spectral bulk is significantly reduced in layers closer to the output. However, the number of outliers to the left of the spectrum in rectangular matrices is slightly increased in later layers, with a peak in layer 19 in our example.

| Layer | Query | Key | Value | Att.-Out. | Up-Proj. | Gate-Proj. | Down-Proj. | Sum |
|---|---|---|---|---|---|---|---|---|
| Layer 0 | [0 \| 1485] | [356 \| 412] | [113 \| 152] | [0 \| 723] | [75 \| 189] | [89 \| 409] | [90 \| 153] | [723 \| 3523] |
| Layer 4 | [0 \| 670] | [211 \| 261] | [111 \| 80] | [0 \| 524] | [69 \| 78] | [96 \| 294] | [161 \| 221] | [648 \| 2128] |
| Layer 9 | [0 \| 769] | [219 \| 288] | [119 \| 199] | [0 \| 663] | [120 \| 253] | [197 \| 470] | [241 \| 347] | [896 \| 2989] |
| Layer 14 | [0 \| 796] | [217 \| 258] | [105 \| 121] | [0 \| 671] | [117 \| 210] | [216 \| 450] | [196 \| 245] | [851 \| 2751] |
| Layer 19 | [0 \| 526] | [166 \| 221] | [131 \| 91] | [0 \| 415] | [184 \| 168] | [205 \| 311] | [183 \| 172] | [869 \| 1904] |
| Layer 24 | [0 \| 484] | [220 \| 230] | [139 \| 114] | [0 \| 259] | [172 \| 185] | [183 \| 266] | [206 \| 226] | [920 \| 1764] |
| Layer 29 | [0 \| 375] | [148 \| 180] | [39 \| 32] | [0 \| 384] | [93 \| 205] | [106 \| 284] | [155 \| 212] | [541 \| 1672] |

Table 3: Singular value outlier count for Llama3-8B in the format [number of left outliers | number of right outliers] with layer resolution.

To also provide a metric on the layer-specific importance of the smallest singular values, we remove deciles from all matrices in that specific layer and measure the perplexity on the BookCorpus dataset to see whether a trend emerges. The base perplexity without removal is 6.0045, and the results are presented in Table 4. We see that removing singular values from a single layer has only a small effect on perplexity in general, a notable exception being the removal of the largest singular values from Layer 0. The second most important decile of Layer 0 is that of the smallest singular values. For the other layers studied, removal of the largest singular values has the largest effect on perplexity, in good agreement with the number of outliers in the first layer. Furthermore, layers 19 and 24 appear to be the least important ones with respect to perplexity, which is also in good agreement with the number of large outliers.

| Layer | Dec. 1 | Dec. 2 | Dec. 3 | Dec. 9 | Dec. 10 |
|---|---|---|---|---|---|
| Layer 0 | 6.3161 | 6.0259 | 6.0303 | 6.2138 | 62355.6 |
| Layer 4 | 6.0405 | 6.0350 | 6.0308 | 6.1828 | 7.3107 |
| Layer 9 | 6.0667 | 6.0284 | 6.0329 | 6.1943 | 6.8791 |
| Layer 14 | 6.0507 | 6.0371 | 6.0349 | 6.1929 | 6.5998 |
| Layer 19 | 6.0214 | 6.0242 | 6.0318 | 6.1359 | 6.4036 |
| Layer 24 | 6.0373 | 6.0317 | 6.0382 | 6.1070 | 6.3084 |
| Layer 29 | 6.1321 | 6.0629 | 6.0508 | 6.1390 | 6.7912 |

Table 4: Effect of the removal of singular value deciles from Llama3-8B on the perplexity score, computed for the BookCorpus dataset.

## H   Scaling Relations

To analyze the scale on which our results for small singular values hold, we compare the scaling of the number of outliers for various model sizes. In general, we know that larger weight matrices display less finite-size effects when comparing them to random matrix theory results derived for infinitely large matrices. To analyze whether there might be a finite size scaling, we compare our three models: Bert with 110M parameters (embedding dimension $d = 768$), Pythia with 506M parameters and $d = 1024$, and Llama with 8.03B parameters and $d = 4096$.

To quantify the number of singular value outliers, we considered the percentage of singular values smaller than the lower Marchenko-Pastur bound (averaged over all layers) in the matrix types where we showed that small singular values are important: For the Up-Projection matrices we find for Bert,

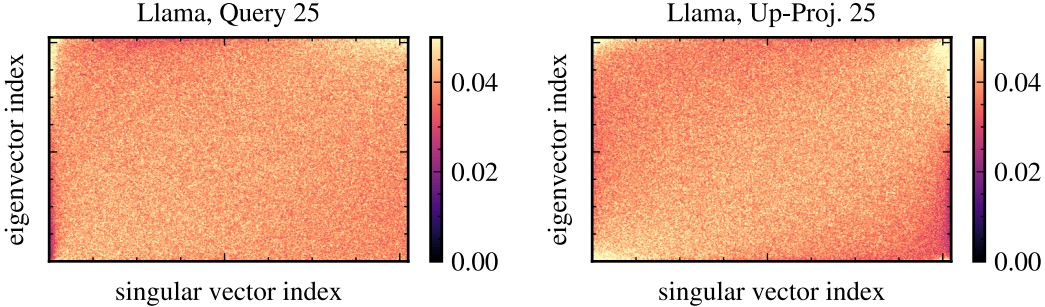

Figure 15: Cosine similarity, $v_k \cdot e_j$, between weight singular vectors $v_k$ and activation covariance eigenvectors $e_j$ for the Query matrix of layer 25 of the Llama model (left panel) and for the Up-Projection matrix of the same layer (right panel). The vectors are rank ordered with respect to the eigenvalues (singular values) such that vectors for the largest values correspond to small indices, i.e. to the bottom (left) of the $y$ ($x$) axis.

Pythia, and Llama $[3.8\%, 5.9\%, 3.2\%]$, and for the Down-Projection matrix $[2.1\%, 6.5\%, 4.7\%]$. We do not observe a clear trend here, which might indicate that the number of relevant small singular values scales linearly with model size, i.e., is a constant in terms of percentages. However, it is challenging to derive scaling relations from only three models, especially since they are far from identical, and other details of training may be relevant too.

# I   Reduction in the Space of Activations

In the main text, we address the relevance of small singular values and demonstrate a correspondence between their corresponding singular vectors and the eigenvectors of the activation covariance matrix. As modern reduction algorithms often aim to reduce the number of network parameters by reducing the dimension of the activation space [3, 40], it is also crucial to identify exactly to which eigenvectors these singular vectors correspond. This would provide additional information about the relevance of specific directions and, therefore, may enable the construction of more precise reduction algorithms.

While a detailed analysis would exceed the scope of this paper, Figure 15 provides an idea of what insights may be gained from such an analysis. Here, small indices (bottom left corner) correspond to eigenvectors ($y$ axis) and singular vectors ($x$ axis) for the largest eigenvalues and singular values, respectively. We observe that in the case of the Query matrix of Llama (left panel), the largest overlaps can be found between the eigenvectors corresponding to the smallest eigenvalues and singular vectors corresponding to the largest singular values, which is surprising as it indicates the relevance of directions in *activation space* which are small in terms of the variance that they carry. A similar observation can be found for the Up-Projection matrix of Llama (right panel), where both the eigenvectors for smallest and largest eigenvalues have an increased overlap with the singular vectors for the largest singular values. Surprisingly, there is a correspondence between vectors for the smallest singular values and smallest eigenvalues, which may indicate that they are very far from random noise. However, further investigations with a wider scope are necessary to provide a clear picture of this phenomenon.

# J   Additional Results on HumanEval

When removing deciles of singular values (starting with the smallest to the largest ones) from the Down-Projection matrix (Down-Proj.), the Attention-Output matrix (Att.-Out.), the Gate-Projection matrix (Gate-Proj.), or the Query matrix, we observe the performances displayed in Table 5 on the HumanEval benchmark. In excellent agreement with previous results, we find that the smallest singular values of the Down-Projection and Gate-Projection matrices are important, which is not the case for the smallest singular values of the quadratic matrices.

Table 5: Effect of removing deciles of singular values from various weight matrices of Llama3-8B on the HumanEval benchmark accuracy. Decile 1 corresponds to the smallest, and Decile 10 to the largest singular values. The full model reaches 32.32% accuracy. Removing the smallest singular values from the Down-Projection and Gate-Projection matrices causes a large drop in performance, confirming their importance. For the quadratic matrices Attention-Output and Query, the smallest singular values appear less critical.

| Matrix | Dec. 1 | Dec. 2 | Dec. 3 | Dec. 4 | Dec. 5 | Dec. 6 | Dec. 7 | Dec. 8 | Dec. 9 | Dec. 10 |
|---|---|---|---|---|---|---|---|---|---|---|
| Down-Proj. | 0.0% | 29.9% | 27.4% | 20.7% | 20.1% | 18.3% | 17.7% | 1.2% | 0.0% | 0.0% |
| Att.-Out. | 34.8% | 32.3% | 34.1% | 30.5% | 36.0% | 29.9% | 30.5% | 28.0% | 29.9% | 0.0% |
| Gate-Proj. | 28.7% | 29.3% | 36.0% | 27.4% | 29.3% | 32.3% | 29.3% | 23.8% | 20.7% | 0.0% |
| Query | 32.3% | 32.3% | 31.7% | 33.5% | 31.1% | 33.5% | 32.9% | 34.8% | 30.5% | 0.0% |

