# OpenReview forum: "Small Singular Values Matter: A Random Matrix Analysis of  Transformer Models"
_NeurIPS.cc/2025/Conference — NeurIPS 2025 poster_

### Official Review · Reviewer_iFU6 · 2025-06-29

**Clarity:** 3
**Significance:** 3
**Originality:** 3
**Rating:** 4
**Confidence:** 5

**Summary:**

This paper uses the MP law to estimate what a typical distribution of singular values should look like. For trained matrices, outliers arise from the typical distribution, on both left and right side (small and large values) for non-square matrices. This reveals that there exist some small singular values that are of relevance to the training of the matrix - thus truncating singular values by magnitude (as is usually done for frobenius norm minimization, per Ben-Eckart) may not be a good idea.

Using activation covariance eigenvectors, correlations are measured with weight singular vectors. It turns out that indeed, singular vectors associated with those singular values falling outside the MP law do indeed excite activation eigenvectors.

**Questions:**

- It is interesting to study RMT applied to weight matrices. But what about activations, there are much more redundancies in activations which can be better exploited to compress a model using matrix factorization, as was shown in [1]. Can we perform a similar study for activations?
[1]:
Sakr, Charbel, and Brucek Khailany. "ESPACE: Dimensionality Reduction of Activations for Model Compression." Neurips 2024

- For the spectral overlaps in Fig 2 & 3, it seems rather qualitative, and more like the authors are trying to to convince us that something is there by telling us it's there (namely by switching between blue and red colors). However, it does seem like some vectors in the blue region do exhibit strong overlap with the activation eigenvectors. I wonder if there is a more proper way to visualize these (maybe log plots). And please comment on why some blue singular vectors have such strong overlap.

- In the experimental setup of section 5, why are deciles used? Why can't we just do the truncation as per the description of the earlier section, i.e., take out the singular values that supposedly arise from the randomness, and leave the outliers that supposedly came about during training? Such a 1-to-1 experiment correlating with the earlier theory would be interesting.


- Not a question but the last sentence of the abstract is repeated twice.

**Ethical Concerns:**

["NO or VERY MINOR ethics concerns only"]

**Final Justification:**

The authors have provided qualitative replies to my questions. I maintain my rating of borderline accept.

**Limitations:**

Yes, it's there on pages 8-9 just before the conclusion.

**Paper Formatting Concerns:**

No formatting issues but I wonder if the plots can be made more lightweight (as in save some bytes). My PDF reader (on Chrome/Windows) glitches a bit when I scroll through the figures.

**Quality:**

3

**Strengths And Weaknesses:**

Nice paper giving a fresh perspective into singular values decomposition of weights. I like the departure from the usual frobenius norm minimization and the fact that the study compares trained matrices to random ones and essentially asks what different because such differences are likely what makes the model strong. Good introduction of interesting concepts.

In terms of weaknesses, there are two main issues:
1) the paper does not showcase empirically how strong the proposed method derived from the study can be at compressing models. It would be good to see comparisons of the method to SOTA techniques in tensor decomposition such as ESPACE, BLAST, etc..
2) Some minor issues that need answering. Please refer to the "Questions" part of the review.

---

> ### Author Rebuttal · Authors · 2025-07-27
>
> We thank the reviewer for their thoughtful and positive report. We appreciate the reviewer's valuable feedback and suggestions. In the following, we answer the reviewer's questions in detail:
>
> **1)** Relevance of eigenvectors corresponding to small eigenvalues in the activation covariance matrix: We thank the reviewer for pointing out this important topic. We have analyzed eigenvalue spectra of activation covariance matrices, but not included them in the manuscript. As in the case of weight matrices, we found outliers to the left of the spectrum, i.e., eigenvalues that are smaller than the bulk. Furthermore, we observed that the singular vectors of small singular values of weight matrices often correspond to eigenvectors of the activation covariance matrix with small eigenvalues.
> This matching could indicate that a similar analysis for the activations could indeed be relevant to compressing algorithms that rely on eigenvalues of the activation covariance matrix.
>
> In [Ashkboos et al., SliceGPT: Compress Large Language Models by Deleting Rows and Columns, ICLR (2024)], the authors project into a lower-dimensional activation space spanned only by the directions of large eigenvectors of the activation covariance matrix, which could be non-optimal in our understanding. The mentioned paper [1] improves upon this by letting the network itself choose the important directions during training by having a static projection matrix. However, the mentioned method [1] cannot reduce the size of already trained networks. In cases like [Ashkboos et al., ICLR (2024)], our results may hence add an important piece of information and may lead to increased performance of pruned networks.
>
> We are happy to add a plot in the final version of the manuscript that illustrates this relation of small singular vectors to small eigenvectors of the activation covariance matrix.
>
> **2)** We thank the reviewer for their question and would like to argue that the results shown in Figures 2 and 3 do go beyond a pure visualization: Figure 2 and Figure 3 display the maximum overlap $\max_k (\textbf{s}_i \cdot \textbf{e}_k )$ for each singular vector $\textbf{s}_i$ with the eigenvectors $\textbf{e}_k$ of the activation covariance matrix, which we interpret as a singular vector to feature correspondence. To give meaning to these plots, we indicate the 3-sigma interval, e.g., the interval in which we expect 99.7\% of the data to fall, if the singular vectors were random. In Figure 3, we observe that the number of blue dots outside this region is indeed in a similar range as statistically expected for a completely random process (2-12 are statistically expected depending on the number of singular vectors, i.e., the matrix size).
> For the red dots, around 2/3 of the datapoints fall outside of this region, which is statistically highly significant, i.e., cannot be explained by a random chance.
>
> In the case of Figure 2, we find seven blue dots outside the region for Bert, which is more than the statistically expected 2.3 dots. We interpret these deviations as a sign that even in the bulk part, we may find some learned information. However, the ratio of statistically significant points to the number of total points is extremely low when compared to the singular values outside of the spectrum.  For Pythia and Llama, the number of outliers is again in the statistically expected range of ~3 for Pythia and ~12 for Llama.
>
> We agree with the reviewer that a log scale could have been chosen, but we tested both and found that a linear scale provides a cleaner view, considering that the range of displayed values only spans roughly one order of magnitude (0.03-0.3).
>
>
> **3)** We agree with the reviewer that other test designs could have been chosen, but we chose our design due to the following reasons: The current setup of removing deciles is easy to understand, and it clearly shows the point that small singular values are more important than larger ones in the case of non-square matrices, as we compare the removal of sets of equal size.
>
> More importantly, in an exploratory study, we have found that removing all singular values that we classify as random (keeping only the outliers to the left and right of the Marchenko-Pastur bulk) would lead to a significant reduction of network performance. One can ask the question: why does the performance decrease if one only removes randomness? This can happen if the network is not fully in the feature learning regime. One can demonstrate this effect in a toy model: when training a network with two hidden layers, with the weights between the input and the first hidden layer completely random and frozen during training. Since the trained weights depend on the random transformation of inputs due to the frozen and random layer, removing singular values from this random layer reduces the network performance significantly. For this reason, a pruning scheme based on the findings presented in our manuscript would need to be more involved: one would need to distinguish between weight matrices close to lazy training (e.g., with only a few outliers) and weight matrices in the feature learning regime. Due to these complications, we have decided not to include pruning experiments in the present manuscript. Nevertheless, we agree with the reviewer that further experiments can be conducted that investigate the phenomenon further.
>
>
> **4)** We thank the reviewer for pointing out the duplicated sentence and will remove it in a revised version.
>
>
> We believe we have thoroughly addressed all of your concerns, and would be grateful if you could let us know in case anything remains unclear or if further clarification is needed.

---

### Official Review · Reviewer_2hdo · 2025-06-30

**Clarity:** 4
**Significance:** 3
**Originality:** 3
**Rating:** 5
**Confidence:** 3

**Summary:**

This paper revisits transformer models via random matrix theory. Empirically, they found out that while small singular values typically represents "outliers", removing those information significantly impacted model performance, even more so than removing bulk values.
After fine-tuning, the bottom decile can become the third–most influential spectral region.
To explain this, the paper proposes a linear teacher–student RMT model in which noise covariance induces small‐value outliers whose vectors carry more task‐relevant information the further they lie from the bulk.

**Questions:**

1. Can the authors provide further explanation on the connection to lazy training? (Appendix D) As to my knowledge, lazy training only seems to happen when one has infinitely wide networks.
2. How does the phenomenon scale with model size? Are small-value outliers equally important in different sizes of models?

**Ethical Concerns:**

["NO or VERY MINOR ethics concerns only"]

**Final Justification:**

The authors have addressed my concerns, I remain my score.

**Limitations:**

Yes, it is right before the conclusion.

**Paper Formatting Concerns:**

The format looks fine to me.

**Quality:**

3

**Strengths And Weaknesses:**

## Strength
1. I think the insights from this paper is novel. This paper highlights an overlooked spectral regime (small singular values) as carriers of learned features, overturning the conventional emphasis on large‐value outliers alone.
2. I think the paper carries out several interesting potential directions in transformer theories, like a new perspective of RMT, analyzing small singular values etc.
3. While the paper mainly present empirical findings, I do think it provides practical guidance, theoretical implications to the ML community.
4. The paper is very well written, easy to follow for readers.

## Weakness
1. To my knowledge, the RMT model used in the paper assumes the inputs to follow Gaussian distribution, and with linear attention. The latter assumption does not seem to be general to all transformer variants.
2. It would be nice to see some further explanation on the connection to "lazy training". So far, it is not clear to me why "lazy training" would lead to their empirical results.

---

> ### Author Rebuttal · Authors · 2025-07-27
>
> We thank the reviewer for their thoughtful and positive report. We appreciate the reviewer's valuable feedback and suggestions. In the following, we answer the reviewer's questions in detail:
>
> **1.** In the lazy training regime, the weights change only very little during training, such that the model depends linearly on the weights around the initialization. This phenomenon is therefore often studied in the Neural Tangent Kernel (NTK) regime: The reviewer is right that this regime emerges naturally in the infinite width limit. However, the lazy regime can also be realized in finite networks.
>
>
> One can manually tune to the lazy regime by rescaling the input of the softmax function in the final layer by a constant $\alpha>1$
> $$
> a_L = \text{softmax} \left( \alpha ( W_L a_{L-1} + b_L ) \right) \ ,
> $$
> while also scaling down the loss by a factor $\( 1/\alpha^2 \)$:
> $$
> l(W, b) = -\frac{1}{N \alpha^2} \sum_{k=1}^N y^{(k)} \cdot \ln\left( a_{\rm out}^{(k)} \right)
> $$
> (see [Chizat et al., NeurIPS 32 (2019)]).
> These changes have the effect that small updates to the weights can have a large impact on the output activations. Additionally, it shows that the learning regime can be tuned continuously from feature to lazy learning.
>
> Depending on model architecture, training data, and many more parameters, different layers of a model may effectively be trained in different learning regimes, even when not manually applying a rescaling. For example, in simple MLP networks on Cifar-10, layers closer to the output tend to be more lazy than layers closer to the input. The learning regime of a weight matrix can be revealed by an RMT analysis, as weights close to initialization follow RMT predictions, such as the Marchenko-Pastur distribution, more closely. This is what we discuss in Appendix D and Fig. 14.
>
> **2.** Scaling relations: When going to larger model sizes, we typically see larger weight matrices that display less finite-size effects. To analyze whether there might be a size scaling, we compare our three models, Bert with 110M parameters (embedding dimension $d=768$), Pythia with 506M parameters and $d=1024$, and Llama with 8.03B parameters and $d=4096$. A first observation is that for all three models, the small singular values in the Up-Projection and Down-Projection layers are important.
>
> To quantify the number of outliers, we considered the percentage of singular values smaller than the lower Marchenko-Pastur bound (averaged over all layers):
> - Bert: Up-Proj:  $3.8\\%$, Down-Proj: $2.1\\%$
> - Pythia: Up-Proj: $5.9\\%$, Down-Proj: $6.5\\%$
> - Llama: Up-Proj: $3.2\\%$, Down-Proj: $4.7\\%$
>
> We do not observe a clear trend here, which might indicate that the number of relevant small singular values scales linearly with model size, i.e., stays identical in terms of percentages.
>
> For Pythia and Llama, we consider the model's perplexity on WikiText. Quantitatively, removing the decile with the smallest singular values (decile 1) results in a comparable perplexity increase relative to the perplexity increase when removing the second largest decile (decile 9):
> - Pythia: Up-Proj: $\Delta PPL(1) / \Delta PPL(9) = 0.72$, Down-Proj: $0.88$
> - Llama: Up-Proj: $\Delta PPL(1) / \Delta PPL(9) = 0.89$, Down-Proj: $1.02$
>
> This suggests that the relevance stays approximately constant, similar to the percentage outside of the Marchenko-Pastur distribution. However, it is challenging to derive scaling relations from only three models, especially since they are far from identical, and other parameters of training might come into play. We will add these numbers to the appendix with a remark that the results show no clear trend, which indicates that this phenomenon persists over several model scales.

---

### Official Review · Reviewer_XvMa · 2025-07-01

**Clarity:** 3
**Significance:** 3
**Originality:** 3
**Rating:** 4
**Confidence:** 2

**Summary:**

By analyzing weight matrices through Random Matrix Theory, the study finds that both the largest and smallest singular values deviate from randomness, indicating they encode meaningful information. Removing non-random singular values greatly impacts model performance, highlighting the overlooked importance of small singular values and offering practical insights for compressing large language models.

**Questions:**

1) I noticed that since small singular values have a significant impact, I speculate that amplifying them will be beneficial for the model's learning, and this seems to be related to the muon optimizer [1]. What is your opinion on this?

2) About evaluation, multiple choice and PPLs are often misleading, and some studies have revealed that many components in the network contribute very little to multiple choice questions[2].  The impact on generating questions is quite different [3].  Therefore, in addition to needle in a haystack, I also suggest that the author measure more benchmarks for generative tasks.




[1] https://kellerjordan.github.io/posts/muon/

[2] Understanding the skill gap in recurrent language models: The role of the gather-and-aggregate mechanism.

[3] Shortgpt: Layers in large language models are more redundant than you expect.

**Ethical Concerns:**

["NO or VERY MINOR ethics concerns only"]

**Final Justification:**

I am inclined to accept, but I hope to include evaluations related to long-term contextual abilities.

**Limitations:**

See weakness.

**Paper Formatting Concerns:**

No.

**Quality:**

3

**Strengths And Weaknesses:**

**Strengths**

1. Theoretical analysis is solid. The paper explain the phenomenon that small singular values that occur in practice also have a significant impact.

2.  It has certain practical significance and provides insight for designing a more suitable pruning or LLM optimization scheme in the future.

3. The article is well presented and easy to read.




**Weakness**
1.  Evaluation is limited. We know that PPL is often inaccurate, especially in long context tasks [1][2], so I strongly recommend the authors to evaluate tasks like needle in a haystack.

[1] Can perplexity reflect large language model’s ability in long text understanding?  ICLR 2024.

[2] Base of RoPE Bounds Context Length. NeurlPS 2024.

---

> ### Author Rebuttal · Authors · 2025-07-27
>
> We thank the reviewer for their thoughtful and positive report. We appreciate the reviewer's valuable feedback and suggestions. In the following, we answer the reviewer's questions in detail:
>
> **1)** The reviewer pointed out another potential application where our theoretical approach to small singular values could be fruitful. To our understanding, the success of Adam compared to SDG with momentum lies mainly in rescaling the update size based on variances, allowing for faster learning along almost flat directions. However, Adam is initialized without amplification and needs a number of update steps to find flat but relevant directions. Under the assumption that small singular values of the gradient matrix are more relevant than their size suggests (something we did not test but appears plausible, considering our results), Muon would then effectively increase the learning speed in these important directions from the start, where Adam takes a long time to figure them out. This could explain the success of Muon in speed learning scenarios.
>
>
> **2)** We agree with the reviewer that evaluation tasks like 'needle in a haystack' could further improve the analysis. Considering that current implementations of this task are testing modern models via their respective APIs, we did not implement this task for our models. Analyzing such commercially available models would be interesting; however, for our analysis to work, we need access to the weights of the model and need to modify them. If the reviewer is aware of a hugginface model-compatible implementation of this task, we would gladly include it.
>
> To strengthen our analysis, we conducted additional experiments on the GSM-8K benchmark and HumanEval for Llama3-8B to enhance the empirical verification of our findings.
> In the spirit of our previous experiments, we remove deciles of singular values from all matrices of a specific type while testing the ability of Llama to solve basic math problems. Under 3-shot prompting and without any removal, Llama 3 reaches an accuracy of 43.2% on the GSM8K benchmark in our settings.
>
> When removing deciles of singular values (starting with the smallest singular values to the larger ones) from the Down-Projection matrix (Down-Proj.), the Attention-Output matrix (Att.-Out.), the Gate-Projection matrix (Gate-Proj.), or the Query matrix, we observe the following performance on GSM-8K:
>
>
> | Llama3.1-8B   | Dec. 1 | Dec. 2 | Dec. 3 | Dec. 4 | Dec. 5 | Dec. 6 | Dec. 7 | Dec. 8 | Dec. 9 | Dec. 10 |
> |-----------|--------|--------|--------|--------|--------|--------|--------|--------|--------|---------|
> | Down-Proj. | 2%    | 30.2% | 28%   | 27.1% | 26.2% | 22%   | 11%   | 15%   | 5%    | 0%    |
> | Att.-Out.  | 40%   | 41.8  | 39%   | 38.8% | 39.8% | 37.1% | 37.6% | 31.4% | 25.1% | 0%    |
> | Gate-Proj. | 34.1% | 37%   | 39.1% | 39.6% | 38.1% | 38%   | 35.9% | 33.3% | 23.6% | 0%    |
> | Query      | 40.3% | 44.3% | 41.4% | 40.5% | 39.5% | 40.1% | 40.7% | 40.7% | 35.8% | 0%    |
>
> Here, decile one corresponds to the smallest and decile ten to the largest singular values. For the Down-Projection matrix, the results show that the smallest singular values (Dec. 1) are the second most important decile, which is in excellent agreement with our previous results for this rectangular matrix (compare Fig. 5 in the manuscript). Looking at the quadratic Attention-Output matrix, the smallest singular values are one of the least important deciles, in agreement with theoretical expectations and previous results.
>
> The smallest singular values of the non-quadratic Gate-Projection matrix are the fourth important decile, in excellent agreement with the results of Fig. 5. For the quadratic Query matrix, only the two largest deciles appear to be important.
>
> A very similar pattern can be observed if we evaluate the capabilities of Llama on the code generation benchmark HumanEval, where the full model scores 32.32% in our settings.
>
> | Llama3.1-8B   | Dec. 1 | Dec. 2 | Dec. 3 | Dec. 4 | Dec. 5 | Dec. 6 | Dec. 7 | Dec. 8 | Dec. 9 | Dec. 10 |
> |--------------|--------|--------|--------|--------|--------|--------|--------|--------|--------|---------|
> | Down-Proj.   | 0.0%   | 29.9%  | 27.4%  | 20.7%  | 20.1%  | 18.3%  | 17.7%  | 1.2%   | 0.0%   | 0.0%    |
> | Att.-Out.    | 34.8%  | 32.3%  | 34.1%  | 30.5%  | 36.0%  | 29.9%  | 30.5%  | 28.0%  | 29.9%  | 0.0%    |
> | Gate-Proj.   | 28.7%  | 29.3%  | 36.0%  | 27.4%  | 29.3%  | 32.3%  | 29.3%  | 23.8%  | 20.7%  | 0.0%    |
> | Query        | 32.3%  | 32.3%  | 31.7%  | 33.5%  | 31.1%  | 33.5%  | 32.9%  | 34.8%  | 30.5%  | 0.0%    |
>
> We again find that the smallest singular values of the Down-Projection and  Gate-Projection matrix are surprisingly important, which is not the case for the quadratic matrices.
>
> We suggest to add these results to the appendix of the manuscript, together with results for the remaining matrix types (once they are ready). We thank the reviewer for their suggestion to improve our manuscript with the inclusion of this additional benchmark.
>
>
> We believe we have thoroughly addressed all of your concerns, and would be grateful if you could let us know in case anything remains unclear or if further clarification is needed.

---

> > ### Comment · Reviewer_XvMa · 2025-08-04
> >
> > Thanks for the author's response.  My concern has been largely resolved. I think the author can consider using the following link [1] for the implementation of "needle in a haystack" and some related tasks about long context.
> >
> >
> > [1] https://github.com/NVIDIA/RULER

---

> > > ### Author Response · Authors · 2025-08-06
> > >
> > > We thank the reviewer for the link. We started the computation of results, however, due to the computational complexity of the task, we cannot provide the results on the requested benchmark in the short time period. We apologize and hope that the two added benchmarks, consisting of code generation and math problems, together with the already existing ones, provide enough empirical evidence for our model.

---

> > > > ### Comment · Reviewer_XvMa · 2025-08-07
> > > >
> > > > I understand that the author was unable to conduct the relevant experiments in a short period of time. But I hope that in the final version, evaluations related to long context capabilities can be taken into consideration. Because this is the one of the fundamental differences between transformer and many other structures, such as LSTM and linear transformer.

---

> > > > > ### Author Response · Authors · 2025-08-07
> > > > >
> > > > > We agree with the reviewer that the ability of LLMs to process long contexts is one of their key properties. We are therefore continuing our simulations on the "needle in a haystack" task and are confident that the results will be ready for inclusion in a potentially final version of the manuscript.

---

### Official Review · Reviewer_7D7J · 2025-07-02

**Clarity:** 3
**Significance:** 3
**Originality:** 3
**Rating:** 4
**Confidence:** 3

**Summary:**

The paper applies Random Matrix Theory to pretrained Transformer weights, showing that both large and small singular-value outliers—identified against the Marchenko–Pastur baseline—align with salient activation directions and materially affect performance. Experiments on BERT, Pythia and Llama plus a teacher-student RMT model support these claims and highlight the oft-overlooked importance of small singular values for pruning and fine-tuning.

**Questions:**

No major concerns at this stage.

**Ethical Concerns:**

["NO or VERY MINOR ethics concerns only"]

**Final Justification:**

Most of my concerns have been adequately addressed, and I appreciate the clarifications and new results. I will maintain my original score.

**Limitations:**

Yes

**Paper Formatting Concerns:**

Not obvious formatting issues observed

**Quality:**

3

**Strengths And Weaknesses:**

### Strengths

1. The paper proposes a novel and interpretable use of the Marchenko–Pastur law as a random baseline to dissect learned signal (outliers) from noise (bulk) in the weight spectra of LLMs, and supports it with overlap ( $O_k$ ) measures and pruning experiments.
2. It rigorously distinguishes between square and rectangular matrices—showing how only the latter can exhibit small singular value outliers—and explains this asymmetry with a rank-1 RMT teacher–student model in Section 6.

### Weaknesses

1. The core experiments are conducted on older models (e.g., BERT, Pythia), which limits the relevance of the findings. Replacing these with more recent state-of-the-art models such as Qwen 2.5/3 would make the results significantly more meaningful.
2. The impact of pruning order and fine-tuning is not yet validated on modern models, and results on real-world tasks such as GSM-8K or MT-Bench are missing.
3. The analysis of outlier counts, overlap scores, and pruning sensitivity across layers is limited to selected blocks; more comprehensive layer-wise statistics are needed.
4. The RMT model in Section 6 makes strong simplifying assumptions (e.g., linear two-layer, Gaussian inputs, rank-1 teacher), which may limit generalizability. Its assumptions and scope should be better discussed and tested numerically.

---

> ### Author Rebuttal · Authors · 2025-07-27
>
> We thank the reviewer for their thoughtful and positive report. We appreciate the reviewer's valuable feedback and suggestions.
>
> **1-2)** We agree that enhancing the analysis beyond our currently largest model (Llama3-8B) would be valuable, but such an extension is challenging in the short time period provided for author answers. However, we conducted additional experiments on the GSM-8K benchmark and HumanEval for Llama3-8B to enhance the empirical verification of our findings.
> In the spirit of our previous experiments, we remove deciles of singular values from all matrices of a specific type while testing the ability of Llama to solve basic math problems. Under 3-shot prompting and without any removal, Llama 3 reaches an accuracy of 43.2% on the GSM8K benchmark in our settings.
>
> When removing deciles of singular values (starting with the smallest singular values to the larger ones) from the Down-Projection matrix (Down-Proj.), the Attention-Output matrix (Att.-Out.), the Gate-Projection matrix (Gate-Proj.), or the Query matrix, we observe the following performance on GSM-8K:
>
>
> | Llama3.1-8B   | Dec. 1 | Dec. 2 | Dec. 3 | Dec. 4 | Dec. 5 | Dec. 6 | Dec. 7 | Dec. 8 | Dec. 9 | Dec. 10 |
> |-----------|--------|--------|--------|--------|--------|--------|--------|--------|--------|---------|
> | Down-Proj. | 2%    | 30.2% | 28%   | 27.1% | 26.2% | 22%   | 11%   | 15%   | 5%    | 0%    |
> | Att.-Out.  | 40%   | 41.8  | 39%   | 38.8% | 39.8% | 37.1% | 37.6% | 31.4% | 25.1% | 0%    |
> | Gate-Proj. | 34.1% | 37%   | 39.1% | 39.6% | 38.1% | 38%   | 35.9% | 33.3% | 23.6% | 0%    |
> | Query      | 40.3% | 44.3% | 41.4% | 40.5% | 39.5% | 40.1% | 40.7% | 40.7% | 35.8% | 0%    |
>
> Here, decile one corresponds to the smallest and decile ten to the largest singular values. For the Down-Projection matrix, the results show that the smallest singular values (Dec. 1) are the second most important decile, which is in excellent agreement with our previous results for this rectangular matrix (compare Fig. 5 in the manuscript). Looking at the quadratic Attention-Output matrix, the smallest singular values are one of the least important deciles, in agreement with theoretical expectations and previous results.
>
> The smallest singular values of the non-quadratic Gate-Projection matrix are the fourth important decile, in excellent agreement with the results of Fig. 5. For the quadratic Query matrix, only the two largest deciles appear to be important.
>
> A very similar pattern can be observed if we evaluate the capabilities of Llama on the code generation benchmark HumanEval, where the full model scores 32.32% in our settings:
>
> | Llama3.1-8B   | Dec. 1 | Dec. 2 | Dec. 3 | Dec. 4 | Dec. 5 | Dec. 6 | Dec. 7 | Dec. 8 | Dec. 9 | Dec. 10 |
> |--------------|--------|--------|--------|--------|--------|--------|--------|--------|--------|---------|
> | Down-Proj.   | 0.0%   | 29.9%  | 27.4%  | 20.7%  | 20.1%  | 18.3%  | 17.7%  | 1.2%   | 0.0%   | 0.0%    |
> | Att.-Out.    | 34.8%  | 32.3%  | 34.1%  | 30.5%  | 36.0%  | 29.9%  | 30.5%  | 28.0%  | 29.9%  | 0.0%    |
> | Gate-Proj.   | 28.7%  | 29.3%  | 36.0%  | 27.4%  | 29.3%  | 32.3%  | 29.3%  | 23.8%  | 20.7%  | 0.0%    |
> | Query        | 32.3%  | 32.3%  | 31.7%  | 33.5%  | 31.1%  | 33.5%  | 32.9%  | 34.8%  | 30.5%  | 0.0%    |
>
> We again find that the smallest singular values of the Down-Projection and  Gate-Projection matrix are surprisingly important, which is not the case for the quadratic matrices.
>
> We suggest to add these results to the appendix of the manuscript, together with results for the remaining matrix types. We thank the reviewer for their suggestion to improve our manuscript with the inclusion of this additional benchmark.
>
>
> **3)** We agree with the reviewer that more comprehensive statistics for each layer would be an interesting addition to the paper, as our current analysis does not consider these differences quantitatively, but only qualitatively. As a first step towards this goal, we provide outlier counts for several layers of Llama3-8B in the format [#LeftOutlier | #RightOutliers] :
>
>
> | Layer   | Query        | Key           | Value         | Att.-Out.     | Up-Proj.      | Gate-Proj.     | Down-Proj.     | Sum             |
> |---------|--------------|---------------|---------------|---------------|---------------|----------------|----------------|-----------------|
> | Layer 0 | [0 \| 1485]  | [356 \| 412]  | [113 \| 152]  | [0 \| 723]    | [75 \| 189]   | [89 \| 409]    | [90 \| 153]    | [723 \| 3523]   |
> | Layer 4 | [0 \| 670]   | [211 \| 261]  | [111 \| 80]   | [0 \| 524]    | [69 \| 78]    | [96 \| 294]    | [161 \| 221]   | [648 \| 2128]   |
> | Layer 9 | [0 \| 769]   | [219 \| 288]  | [119 \| 199]  | [0 \| 663]    | [120 \| 253]  | [197 \| 470]   | [241 \| 347]   | [896 \| 2989]   |
> | Layer 14| [0 \| 796]   | [217 \| 258]  | [105 \| 121]  | [0 \| 671]    | [117 \| 210]  | [216 \| 450]   | [196 \| 245]   | [851 \| 2751]   |
> | Layer 19| [0 \| 526]   | [166 \| 221]  | [131 \| 91]   | [0 \| 415]    | [184 \| 168]  | [205 \| 311]   | [183 \| 172]   | [869 \| 1904]   |
> | Layer 24| [0 \| 484]   | [220 \| 230]  | [139 \| 114]  | [0 \| 259]    | [172 \| 185]  | [183 \| 266]   | [206 \| 226]   | [920 \| 1764]   |
> | Layer 29| [0 \| 375]   | [148 \| 180]  | [39 \| 32]    | [0 \| 384]    | [93 \| 205]   | [106 \| 284]   | [155 \| 212]   | [541 \| 1672]   |
>
> It seems that for some matrices related to the attention mechanism (Query, Key, Value, Att.-Output), the number of outliers to the right of the spectral bulk is significantly reduced in layers closer to the output. However, the number of outliers to the left of the spectrum in rectangular matrices slightly increases in later layers, with a peak in layer 19 in our example.
>
> Studies on the overall relation between depth and relevance can be found in the literature, see e.g. [Chen et al., Streamlining redundant layers to compress large language models, ICLR (2025)]. However, the importance of small singular values has not been studied so far. To provide a metric on the importance of the smallest singular values in these layers, we remove deciles from all matrices in that specific layer and measure the perplexity on the BookCorpus dataset to see whether a trend emerges. The base perplexity without removal is 6.0045.
>
> | Layer   | Dec. 1  | Dec. 2  | Dec. 3  | Dec. 9  | Dec. 10 |
> |---------|---------|---------|---------|---------|---------|
> | Layer 0 | 6.3161  | 6.0259  | 6.0303  | 6.2138  | 62355.6 |
> | Layer 4 | 6.0405  | 6.0350  | 6.0308  | 6.1828  | 7.3107  |
> | Layer 9 | 6.0667  | 6.0284  | 6.0329  | 6.1943  | 6.8791  |
> | Layer 14| 6.0507  | 6.0371  | 6.0349  | 6.1929  | 6.5998  |
> | Layer 19| 6.0214  | 6.0242  | 6.0318  | 6.1359  | 6.4036  |
> | Layer 24| 6.0373  | 6.0317  | 6.0382  | 6.1070  | 6.3084  |
> | Layer 29| 6.1321  | 6.0629  | 6.0508  | 6.1390  | 6.7912  |
>
> We see that removing singular values from only one layer has only a small effect on perplexity in general, a notable exception being the removal of the largest singular values from Layer 0. The second most important decile of Layer 0 are the smallest singular values. For the other layers studied, removal of the largest singular values has the largest effect on perplexity,  in good agreement with the number of outliers in the first layer. Furthermore, layers 19 and 24 appear to be the least important ones with respect to perplexity, which is also in good agreement with the number of large outliers.
>
> We suggest adding the above layer-wise results to the appendix.
>
>
> **4)**  We thank the reviewer for pointing out that the limitations of the theoretical model should be discussed more thoroughly in our manuscript. The reviewer is right that our model uses simplifying assumptions.
> Our goal is to provide a minimal model that captures a mechanism for how small outliers can be produced during training. To us, it was initially very counterintuitive that gradient descent would produce these small outliers, as the general wisdom is that they are negligible. Unfortunately, we are unable to provide numerical results for non-linear activations within the time window for author-reviewer discussions.
>
> However, we will add the following paragraph to section 6 about the minimal RMT model:
> The aim of this section is to provide a minimal RMT model for the occurrence of small singular value outliers. For this, we consider a simple two-layer linear student model, trained on Gaussian inputs, and labels produced by a linear rank-1 teacher. The effects of non-trivial correlations in the noise would depend on whether the noise has predominantly small or large covariance in the direction of the outlier.

---

### Note · Authors · 2025-08-12

We thank the reviewers for their thoughtful and informed questions, which we have addressed in detail in our individual responses. We take the reviewers’ quick acknowledgements as an indication that these questions have been satisfactorily resolved.

We would like to highlight some key points that emerged during the discussion:

Several reviewers noted potential applications where our findings could be beneficial.
Reviewer iFU6 connected our research to a recent NeurIPS 2024 paper on reducing LLMs via subspace projection. As discussed, we believe it is possible to identify a subspace even more effective for such reductions.
Reviewer XvMa drew a link to optimizers, where we suggest that our theoretical results on small singular values may help explain why the Muon optimizer outperforms Adam in terms of speed.

We also wish to underscore several positive appraisals from the reviewers:

2hdo:  I think the insights from this paper are novel. I think the paper carries out several interesting potential directions in transformer theories, like a new perspective of RMT, analyzing small singular values etc.
It provides practical guidance, theoretical implications to the ML community.


XvMa: Theoretical analysis is solid. It has certain practical significance and provides insight for designing a more suitable pruning or LLM optimization scheme in the future.

iFU6: Fresh perspective into the singular value decomposition of weights. Good introduction of interesting concepts.

7D7J: The paper proposes a novel and interpretable use of the Marchenko–Pastur law as a random baseline to dissect learned signal (outliers) from noise (bulk).


We greatly appreciate the reviewers’ recognition of both the novelty and practical relevance of our work. We are encouraged by the connections drawn to diverse application areas, where our findings could have a meaningful impact. We hope these positive evaluations will support the acceptance of our manuscript.

---

### Decision · Program_Chairs · 2025-09-17

**Decision:**

Accept (poster)

**Comment:**

This paper analyzes the singular value spectrum of pretrained language models using random matrix theory and finds that both large and small singular values are outliers. The analysis is further validated by removing singular values from different weight matrices from pretrained language models. All reviewers agree that the paper offers novel insights and perspectives, and the authors have successfully addressed most of their concerns during the rebuttal. Therefore, I recommend acceptance in consensus with the reviewers.

The AC suggests that the authors incorporate the reviewers’ suggestions in their camera-ready version, such as adding evaluations on long-term contextual abilities.